# Rethinking Pretraining Data Detection for LLMs: From Local to Global

Chenye Ke [1]   Yan Zhuang [2]   Zirui Liu [1]   Qi Liu [1]

## Abstract

The advancements of Large Language Models (LLMs) are primarily attributed to massive pretraining data, which also introduces risks like privacy leakage and data contamination. Therefore, it is crucial to determine whether an LLM has been trained on a given target text. Existing detection methods primarily rely on local statistics of isolated tokens (e.g., those with the lowest probabilities), neglecting the probability dynamics during the token generation process. In this paper, we shift the detection paradigm from a local token to a global sequence perspective, grounded in the core intuition that memorized sequences exhibit volatility patterns distinct from those generated via inference. We propose Adaptive Entropic Convolutional Analysis (AECA), a framework that conceptualizes the probability sequence as a dynamic signal, integrating calibration with convolutional filtering to effectively capture memorization signals. Extensive experiments demonstrate that AECA surpasses previous methods by up to 1.5% in average AUC on the WikiMIA benchmark, with its advantage being particularly pronounced in long-text scenarios.

## 1. Introduction

In recent years, LLMs have achieved breakthrough advancements across diverse domains (Zhao et al., 2023; Guo et al., 2025; Liu et al., 2026), largely driven by the massive expansion of training data scales (Kaplan et al., 2020). However, model developers increasingly keep the full composition and provenance of their training data undisclosed (e.g., GPT-3 (Brown et al., 2020)). This lack of transparency poses severe risks, such as privacy infringement (Mozes et al., 2023) and, more critically, data contamination (Dong et al.,

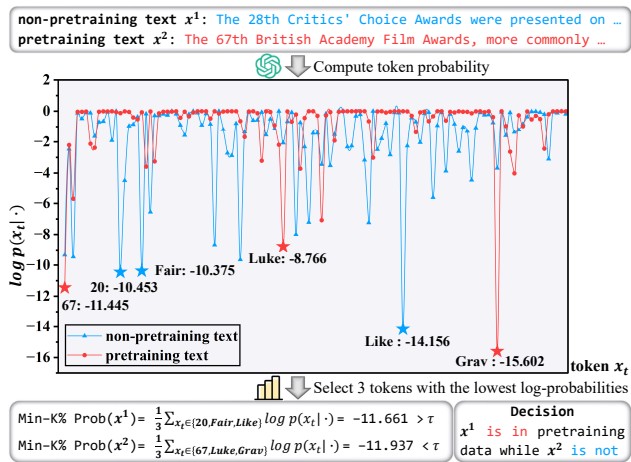

*Figure 1.* **Limitation of the local token perspective.** For non-pretraining text $x^1$ and pretraining text $x^2$, selecting the lowest $k\%$ log-probability tokens (marked with stars) results in a higher average score for $x^1$ than $x^2$. Consequently, local statistics misclassify $x^1$ as pretraining data and $x^2$ as non-pretraining.

2024). The latter undermines scientific evaluation, as the unintentional inclusion of benchmark test data in pretraining corpora compromises the fairness of performance evaluations (Sainz et al., 2023; Zhuang et al., 2025). Consequently, determining whether a specific text was used during the pretraining phase of a target LLM has emerged as a crucial research task.

Pretraining data detection aims to determine whether a specific text belongs to the target model's training corpus. Mainstream research predominantly relies on local statistical metrics, such as Min-K% Prob (Shi et al., 2024), which aggregates the probabilities of the lowest-$k\%$ tokens. Although subsequent studies have introduced calibration mechanisms to mitigate interference from high-frequency words (Zhang et al., 2024; 2025), these approaches essentially rely on static statistics derived from local token subsets. They neglect the probability dynamics during the token generation process and fail to capture the holistic prediction state. This limitation renders them highly sensitive to the hyperparameter $k$. As illustrated in Figure 1, although the non-training sample $x^1$ (blue line) exhibits a generally lower global probability profile, the average log-probability of its three minimum-probability tokens is paradoxically higher than that of the training sample $x^2$ (red line). From a local statistical perspective, this non-training sample would be

[1]State Key Laboratory of Cognitive Intelligence, University of Science and Technology of China [2]College of Artificial Intelligence, Nanjing University of Aeronautics and Astronautics. Correspondence to: Qi Liu <qiliuql@ustc.edu.cn>.

*Proceedings of the 43$^{rd}$ International Conference on Machine Learning*, Seoul, South Korea. PMLR 306, 2026. Copyright 2026 by the author(s).

erroneously classified as training data.

To overcome the limitations of the local token statistics discussed above, we propose shifting the detection paradigm to a global sequence perspective. This shift is grounded in the observation that LLMs demonstrate human-like cognitive abilities, particularly regarding memory (Niu et al., 2024). Specifically, the retrieval of memorized content in LLMs is typically fluent and determinate, whereas reasoning processes are accompanied by uncertainty and fluctuation (Kadavath et al., 2022; Dong et al., 2024). Therefore, memorized content and reasoning-based content exhibit distinct volatility patterns during the token generation process of LLMs. *Based on this intuition, we posit that the key to detection lies in capturing these global volatility features of the probability flow rather than relying solely on local token-level probabilities.*

Guided by this insight, we propose a pretraining data detection method based on a global sequence perspective—**Adaptive Entropic Convolutional Analysis (AECA)**, as shown in Figure 2. Unlike simple statistical aggregation, AECA treats the probability sequence as a dynamic signal to effectively capture the probability volatility throughout the token generation process. Our contributions are as follows:

- We reformulate the LLM pretraining data detection problem by shifting from local token statistics to a global sequence perspective, leveraging distinct volatility patterns to distinguish mechanical memorization from reasoning-based generation.

- We propose a novel framework that integrates self-information calibration with convolutional filtering to effectively capture global volatility patterns missed by local statistical methods.

- Extensive experiments demonstrate that this framework achieves superior performance across mainstream benchmarks, outperforming existing baselines by up to 1.5% in average AUC on WikiMIA, with particularly significant advantages in long-text scenarios.

## 2. Related Work

**Membership Inference Attacks.** Membership Inference Attacks (MIA) were originally proposed by Shokri et al. (2017) with the aim of determining whether a given data sample belongs to the target model's training set by exploiting overfitting (Yeom et al., 2018). As an instrument for assessing privacy risks, MIA was initially applied in the domains of tabular data and computer vision (Carlini et al., 2022a;b; 2023; Zarifzadeh et al., 2024), serving to quantify privacy leakage vulnerabilities and audit the effectiveness of privacy protection mechanisms (Jayaraman & Evans, 2019; Nasr et al., 2021; Steinke et al., 2023). With the rapid ad-

vancement of LLMs, the research focus of MIA has gradually shifted towards generative models. Early research in the NLP domain primarily concentrated on detecting data membership during the fine-tuning stage (Song & Shmatikov, 2019; Hisamoto et al., 2020; Mattern et al., 2023). More recently, the research scope has expanded to the detection of pretraining data. This shift is significant for mitigating legal risks associated with copyright infringement (Meeus et al., 2024) and ensuring evaluation fairness against benchmark contamination (Xu et al., 2024).

**Pretraining data detection for LLMs.** The problem of pretraining data detection can be formalized as a specific instance of MIA applied to LLMs. This is crucial for ensuring the validity of benchmark evaluations and preventing performance overestimation arising from data contamination (Magar & Schwartz, 2022; Liu et al., 2025). Based on model access privileges, existing methods can be categorized into three paradigms: white-box, gray-box, and black-box detection (Cheng et al., 2025). *White-box settings* assume access to internal model parameters or raw corpora. Early studies primarily quantified the extent of overlap by calculating $n$-gram statistics between evaluation samples and the pretraining corpus (Wei et al., 2021; Chowdhery et al., 2023). However, restricted by the parameter and data confidentiality of closed-source models (e.g., GPT-4), such methods face limited application in practical scenarios. *Black-box settings* rely solely on the final text generated by the model. Such methods typically employ specific prompting strategies to induce the model to regurgitate memorized content (Golchin & Surdeanu, 2023; Weller et al., 2024). However, due to high sensitivity to prompt design, their robustness remains limited.

*Gray-box settings* assume access to output probability distributions and represent the core focus of current research. Carlini et al. (2021) established a baseline based on perplexity and introduced calibration techniques to decouple text complexity from memorization effects. Recent studies predominantly rely on aggregated statistics of specific token probabilities, using either raw or calibrated likelihoods of local subsets (e.g., the lowest-$k$% tokens) as detection metrics (Shi et al., 2024; Zhang et al., 2024; 2025). Furthermore, Ye et al. (2024) detect by quantifying the confidence polarization distance under input perturbations. *Nonetheless, these methods tend to treat token probabilities as independent data points, relying on static or local statistics while neglecting the global probability dynamics in the sequence generation process.*

In this paper, we propose AECA, a gray-box method that analyzes the prediction state from a global sequence perspective rather than focusing on isolated statistics of specific tokens. Our results demonstrate that AECA exhibits superior performance to baseline methods.

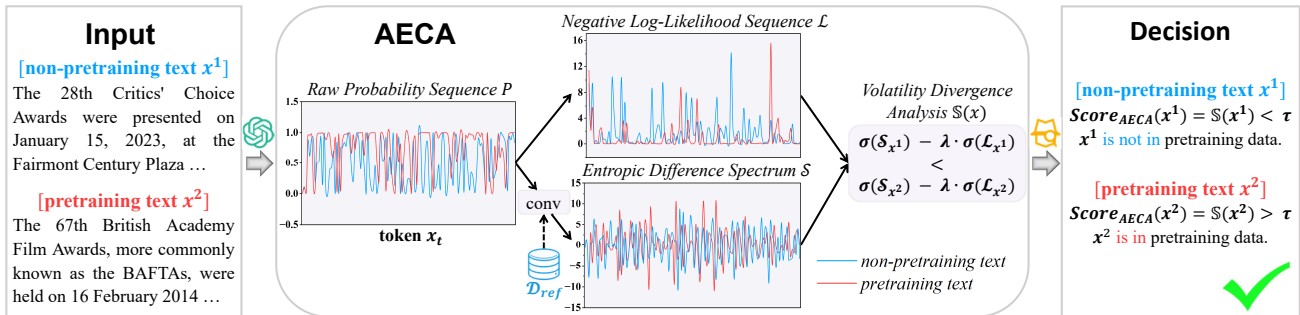

*Figure 2.* **Overview of the AECA framework.** The process consists of four steps: (a) **Probability Acquisition** obtains raw token probabilities from the LLM; (b) **Adaptive Entropic Convolution** utilizes a convolutional filter on calibrated probabilities to highlight fluctuations; (c) **Volatility Divergence Analysis** computes the divergence score between the Entropic Difference Spectrum and the Negative Log-Likelihood Sequence; and (d) **Binary Decision** classifies the text as pretraining data if the divergence exceeds a threshold.

## 3. Problem Definition

Following the definition of MIA (Shokri et al., 2017), given a data instance $x$ and an autoregressive LLM $\mathcal{M}$ trained on a dataset $\mathcal{D}$, the objective is to infer whether $x$ belongs to the training set (i.e., to determine the validity of $x \in \mathcal{D}$). This detection process is typically conducted under a gray-box setting, where the attacker has access only to the model's output statistics (e.g., loss value, logits, and token probabilities) but lacks access to model weights, gradients, or the pretraining dataset (Mattern et al., 2023; Zhang et al., 2025; Cheng et al., 2025).

Mathematically, this problem is formulated as a binary classification task. The core challenge lies in designing a discriminative scoring function $s(x; \mathcal{M})$ and determining a decision threshold $\tau$ to construct a classifier $\mathcal{A}$:

$$\mathcal{A}(x, \mathcal{M}) = \begin{cases} 1 & (x \in \mathcal{D}), \quad s(x; \mathcal{M}) \geq \tau; \\ 0 & (x \notin \mathcal{D}), \quad s(x; \mathcal{M}) < \tau. \end{cases} \quad (1)$$

In this definition, the sample is represented as a sequence of tokens $x = \{x_1, x_2, \ldots, x_T\}$, where each $x_t$ belongs to the vocabulary $\mathcal{V}$ of the model $\mathcal{M}$. The LLMs generate these tokens sequentially based on the conditional probability distribution $p(x_t \mid x_1, \ldots, x_{t-1})$ (we denote $x_1, \ldots, x_{t-1}$ as $x_{<t}$ in the remainder of this paper). Consequently, the efficacy of detection hinges on constructing a robust metric $s(x; \mathcal{M})$ that effectively captures the behavioral discrepancy between training and non-training data (Zhang et al., 2025).

**Traditional Local-Token Perspective Detection.** Existing gray-box detection paradigms identify pretraining samples by aggregating the probabilities of a specific token subset. The scoring metric $s(x; \mathcal{M})$ is formulated as:

$$s(x; \mathcal{M}) = \frac{1}{|E(x)|} \sum_{x_t \in E(x)} F(\log p(x_t | x_{<t})), \quad (2)$$

where $E(x)$ denotes the selected subset of tokens (e.g., the set of tokens with the lowest $k\%$ likelihoods), $|E(x)|$ repre-

sents the cardinality of the subset, and $F(\cdot)$ is a calibration function. However, these methods fundamentally operate by statistically examining tokens in isolation. By restricting the evaluation to a discrete subset $E(x)$, they neglect the integral sequential dependencies and transition dynamics inherent in the sentence. To overcome these limitations, we treat the probability sequence as a holistic signal rather than a collection of isolated points.

## 4. Global Sequence Analysis via AECA

In this section, to operationalize the global perspective, we introduce **A**daptive **E**ntropic **C**onvolutional **A**nalysis **(AECA)**. Unlike traditional methods that rely on local token statistics, AECA derives its detection score from the *fluctuation dynamics* of the entire sequence.

### 4.1. Adaptive Entropic Convolution

We denote the raw probability sequence of the target text $x$ as $\mathcal{P} = \{p_\theta(x_t)\}_{t=1}^T$, where $p_\theta(x_t) = P_\mathcal{M}(x_t \mid x_{<t})$. However, the probability sequence $\mathcal{P}$ is often noisy due to the high frequency of common words (e.g., 'the', 'is'), which naturally yield high confidence regardless of memorization status (Jiang et al., 2019; Zhang et al., 2024; 2025). To mitigate the smoothing effect of common words and capture the true sequential dependency, we design an *Adaptive Entropic Convolution* to transform the raw sequence $\mathcal{P}$ into a fluctuation spectrum by integrating self-information calibration with a high-pass filter.

**Construction of Self-Information.** Following the distribution estimation strategy of (Zhang et al., 2024), we utilize a large-scale general corpus $\mathcal{D}_{\text{ref}}$ to approximate the token frequency $P_{\text{ref}}$ under the natural language background distribution. To address the zero-probability problem caused by data sparsity, we apply *Laplace Smoothing*:

$$P_{\text{ref}}(x_t) = \frac{C(x_t; \mathcal{D}_{\text{ref}}) + \alpha}{N + \alpha|\mathcal{V}|}, \quad (3)$$

where $C(x_t; \mathcal{D}_{\text{ref}})$ denotes the count of token $x_t$ in the reference corpus, $N$ is the total token count, $|\mathcal{V}|$ is the vocabulary size of $\mathcal{M}$, and $\alpha$ is the smoothing coefficient.

Then, we define the *Inherent Self-Information* of a token $x_t$ as $\mathcal{I}_{\text{self}}(x_t) = -\log P_{\text{ref}}(x_t)$. This metric quantifies the objective scarcity of the token within the general distribution (Hale, 2001). Specifically, a higher token frequency corresponds to a lower $\mathcal{I}_{\text{self}}$, indicating reduced scarcity. To capture the interaction between model confidence and this objective scarcity, we establish the following definition:

**Definition 4.1** (Entropic Potential). Given a time step $t$, the Entropic Potential, denoted as $\Phi(t)$, is defined as the coupling of the model's conditional probability and the token's inherent self-information:

$$\Phi(t) \triangleq p_\theta(x_t \mid x_{<t}) \cdot \mathcal{I}_{\text{self}}(x_t). \tag{4}$$

The potential $\Phi(t)$ increases sharply when the model exhibits anomalously high certainty towards inherently rare tokens (i.e., high $\mathcal{I}_{\text{self}}$). This specific deviation from expected long-tail generalization (Feldman, 2020) serves as the primary statistical signal for AECA.

**Entropic Difference Analysis.** Merely relying on static token values is insufficient to capture the volatility of the sequence. To extract fluctuation signals associated with memory triggers, we formalize the entropic difference response as the discrete gradient of $\Phi$:

**Definition 4.2** (Entropic Difference Response). Memory triggers are typically accompanied by drastic local oscillations in Entropic Potential. To capture these oscillations, we apply a 1D discrete convolution with a kernel $\mathbf{k} = [1, -1]$ to the potential sequence $\Phi$. The response value $\mathcal{S}$ at step $t$ is defined as:

$$\begin{aligned} \mathcal{S}[t] = (\Phi * \mathbf{k})[t] &\triangleq \Phi(t) - \Phi(t+1) \\ &= p_\theta(x_t) \cdot \mathcal{I}_{\text{self}}(x_t) - p_\theta(x_{t+1}) \cdot \mathcal{I}_{\text{self}}(x_{t+1}). \end{aligned} \tag{5}$$

This operation generates the *Entropic Difference Spectrum* $\mathcal{S} = \{\mathcal{S}[t]\}_{t=1}^{T-1}$. To theoretically validate its capability in capturing fluctuations, we present the following proposition concerning its spectral properties.

**Proposition 4.3.** *The Entropic Difference Response acts as a high-pass operator sensitive to abrupt variations in Entropic Potential. It theoretically isolates memorization traces by suppressing the smooth, low-frequency components characteristic of natural semantic generation, while selectively amplifying the sharp, high-frequency oscillations induced by overfitting.*

The detailed proof can be found in Appendix A. Through this transformation, smooth natural generation signals are suppressed, while anomalous fluctuations stemming from overfitting-induced memorization are amplified.

---

**Algorithm 1** Adaptive Entropic Convolutional Analysis

**Input:** A token sequence to be detected $x = x_1 x_2 \ldots x_T$, a target LLM $\mathcal{M}$, reference corpus $\mathcal{D}_{\text{ref}}$, smoothing factor $\alpha$, weight $\lambda$, and decision threshold $\tau$.

**Output:** Binary decision $\mathcal{A} \in \{0, 1\}$.

    *// Phase 1: Sequence Metrics Extraction*
1: **for** $t = 1$ **to** $T$ **do**
2:     Get raw probability $\mathcal{P}$: $p_\theta(x_t) \leftarrow P_\mathcal{M}(x_t \mid x_{<t})$.
3:     Compute NLL loss $\mathcal{L}$: $\ell_t \leftarrow -\log p_\theta(x_t)$.
4:     Compute token frequency: $P_{\text{ref}}(x_t) \leftarrow \frac{C(x_t; \mathcal{D}_{\text{ref}})+\alpha}{N+\alpha\mathcal{V}}$.
5:     Compute Self-Info $\mathcal{I}_{\text{self}}$: $\mathcal{I}_{\text{self}}(x_t) \leftarrow -\log P_{\text{ref}}(x_t)$.
6: **end for**
    *// Phase 2: Adaptive Entropic Convolution*
7: **for** $t = 1$ **to** $T - 1$ **do**
8:     $\mathcal{S}[t] \leftarrow \mathcal{P}[t] \cdot \mathcal{I}_{\text{self}}[t] - \mathcal{P}[t+1] \cdot \mathcal{I}_{\text{self}}[t+1]$.
9: **end for**
    *// Phase 3: Volatility Divergence Analysis*
10: Compute Volatility Divergence Score:
    $\mathbf{S}_{\text{AECA}} \leftarrow \sigma(\mathcal{S}) - \lambda \cdot \sigma(\mathcal{L})$.
    *// Phase 4: Binary Decision*
11: **return** 1 **if** $\mathbf{S}_{\text{AECA}}(x) \geq \tau$ **else** 0.

---

## 4.2. Volatility Divergence Analysis

We propose the theory of *Volatility Divergence* to explain the mechanism of AECA. The core intuition is that for memorized samples, the volatility of the loss function tends to collapse, whereas the volatility of the Entropic Difference Spectrum is preserved; conversely, for generalized samples, the volatilities of both are coupled. To capture this divergence, we formally define the detection metric $\mathbf{S}_{\text{AECA}}$ as:

$$\mathbf{S}_{\text{AECA}}(x) = \sigma(\mathcal{S}_x) - \lambda \cdot \sigma(\mathcal{L}_x), \tag{6}$$

where $\sigma(\cdot)$ denotes the standard deviation, $\mathcal{S}_x$ is the Entropic Difference Spectrum, $\mathcal{L}_x = \{-\log p_\theta(x_t)\}$ is the Negative Log-Likelihood (NLL) sequence, and $\lambda$ is a coefficient.

To theoretically justify the effectiveness of this metric, we present the following lemma regarding the statistical behavior of $\mathbf{S}_{\text{AECA}}$ under different model states:

**Lemma 4.4** (Volatility Divergence). *Assume that for a valid text sequence $x$ in the natural language domain, the volatility of its self-information possesses a lower bound $\mu$ (i.e., $\sigma(\mathcal{I}_x) \geq \mu$). Let the Generalization State be $\mathbb{G}$ with fitting error bound $\gamma$, and the Memorization State be $\mathbb{M}$ with an empirical error bound $\epsilon \ll \gamma$. Then, there exists a coefficient $\lambda > \frac{(2\gamma+\epsilon) \log \frac{N+\alpha|\mathcal{V}|}{\alpha} + \log \frac{N+\alpha}{\alpha}}{\mu - (\gamma + \frac{\epsilon}{2})}$ such that:*

$$[\sigma(\mathcal{S}_x) - \lambda\sigma(\mathcal{L}_x)]_\mathbb{M} > [\sigma(\mathcal{S}_x) - \lambda\sigma(\mathcal{L}_x)]_\mathbb{G}. \tag{7}$$

The detailed proof is provided in Appendix B. Guided by this theory, AECA is expected to yield higher response

*Table 1.* Summary Statistics of Benchmarks: Each benchmark comprises a substantial and balanced number of both training and non-training samples. 'Text Length' denotes the number of words within each sample, and '#Examples' represents the total number of samples included.

| Benchmark | Data source | Text length | #Examples | Applicable models |
|---|---|---|---|---|
| WikiMIA | Wikipedia | 32
64
128
256 | 776
542
250
82 | Open-source LLMs
(2017–2023) |
| ArXiv | Pile | $100 \sim 200$ | 2000 | Pythia series |
| HackerNews | Pile | $100 \sim 200$ | 1910 | Pythia series |
| PubMed Central | Pile | $100 \sim 200$ | 2000 | Pythia series |

values on memorized samples, enabling effective discrimination between training and non-training data.

### 4.3. Binary Decision

Finally, we execute the detection by comparing the computed score against a pre-defined threshold $\tau$. Specifically, a sample $x$ is classified as a training member ($x \in \mathcal{D}$) if $\mathbf{S}_{\text{AECA}}(x) \geq \tau$, and as a non-member otherwise. The pseudocode of AECA is presented in Algorithm 1.

## 5. Experimental Settings

**Baseline Methods.** To rigorously evaluate the efficacy of our proposed framework, we benchmark it against eight representative gray-box algorithms which are categorized into two paradigms:

- **Reference-free Methods.** These methods rely solely on raw output probabilities or statistics derived from the target model. ***PPL*** (Yeom et al., 2018; Carlini et al., 2021) uses the absolute sequence perplexity as a metric for memorization. ***Min-K%*** (Shi et al., 2024) calculates the average log-probability of the $k\%$ tokens with the lowest likelihoods. ***Min-K%++*** (Zhang et al., 2025) refines this by incorporating vocabulary-wise normalization. ***PAC*** (Ye et al., 2024) measures the polarization distance of model confidence (specifically contrasting the Max-$k_1\%$ and Min-$k_2\%$ partitions) between the original input and its adjacent samples generated via random swap augmentation.

- **Reference-based Methods.** These methods employ external models or corpora to calibrate the target model's output. ***Lowercase*** (Carlini et al., 2021) compares the perplexity gap between the original and lowercased input. ***Zlib*** (Carlini et al., 2021) normalizes the model's perplexity using the zlib compression entropy of the text. ***Ref*** (Carlini et al., 2021) computes the likelihood ratio between the target and a smaller refer-

ence model. ***DC-PDD*** (Zhang et al., 2024) calibrates the probability distribution by leveraging frequency statistics from a reference corpus.

**Benchmarks.** We focus on two pretraining data detection benchmarks: WikiMIA (Shi et al., 2024) and MIMIR (Duan et al., 2024). **WikiMIA** serves as the pioneering benchmark for pretraining data detection, comprising texts derived from Wikipedia events. The distinction between training and non-training data is determined based on timestamps. WikiMIA specifically categorizes data based on sentence lengths, aiming to facilitate fine-grained evaluation. Furthermore, it incorporates two settings: *original* and *paraphrased* texts[1]. The former assesses the capability to detect verbatim training texts, while the latter utilizes ChatGPT to paraphrase training texts for fuzzy memorization evaluation. **MIMIR** is constructed upon the Pile corpus (Gao et al., 2020), where training and non-training samples are strictly drawn from the training and test splits of the same dataset, respectively. Research indicates that this same-source sampling strategy significantly eliminates distribution shifts and temporal discrepancies, rendering MIMIR more challenging than WikiMIA (Zhang et al., 2025). In our experiments, we selected three representative subsets—*ArXiv, HackerNews, and PubMed Central*—to evaluate detection performance across diverse domains, ranging from technical documentation and social media discussions to medical literature. The statistics of the datasets are shown in Table 1. Further details on their composition can be found in Appendix C.

**Models.** WikiMIA is applicable to a diverse range of models, as Wikipedia dumps are ubiquitously incorporated into the pretraining corpora of numerous LLMs. Specifically, we selected eight models released between 2017 and 2023, including Mamba-1.4B(Gu & Dao, 2024), GPT-Neo-

---

[1] Although the paraphrased settings are introduced in WikiMIA, the corresponding datasets are not currently publicly available. In our experiment, we employed ChatGPT to conduct the paraphrasing, with specific details provided in Appendix C.

*Table 2.* AUC results on the WikiMIA benchmark (Shi et al., 2024). Due to space constraints, we present results for text lengths of 128 and 256, **as longer sequences better demonstrate the advantages of the global perspective**; full results are detailed in Appendix H. *Ori.* and *Para.* denote the original and paraphrased text settings, respectively. Bold values indicate the best results among all methods in each column. The proposed AECA method outperforms existing methods across the majority of settings, achieving observable improvements, particularly in long-text detection.

| Len. | Method | Mamba-1.4B | | GPT-Neo-2.7B | | OPT-6.7B | | Pythia-6.9B | | Pythia-12B | | GPT-NeoX-20B | | Avg. |
|---|---|---|---|---|---|---|---|---|---|---|---|---|---|---|
| | | *Ori.* | *Para.* | *Ori.* | *Para.* | *Ori.* | *Para.* | *Ori.* | *Para.* | *Ori.* | *Para.* | *Ori.* | *Para.* | |
| 128 | PPL | 0.636 | 0.631 | 0.636 | 0.632 | 0.622 | 0.613 | 0.659 | 0.657 | 0.660 | 0.659 | 0.699 | 0.689 | 0.649 |
| | Ref | 0.599 | 0.609 | 0.585 | 0.561 | 0.632 | 0.637 | 0.664 | 0.662 | 0.657 | 0.658 | 0.680 | 0.631 | 0.631 |
| | Lowercase | 0.584 | 0.591 | 0.601 | 0.604 | 0.580 | 0.586 | 0.608 | 0.619 | 0.619 | 0.624 | 0.658 | 0.670 | 0.612 |
| | Zlib | 0.658 | 0.612 | 0.662 | 0.611 | 0.640 | 0.598 | 0.681 | 0.635 | 0.682 | 0.638 | 0.717 | 0.666 | 0.650 |
| | Min-K% | 0.664 | 0.676 | 0.664 | 0.673 | 0.663 | 0.665 | 0.700 | 0.701 | 0.706 | 0.713 | 0.742 | 0.737 | 0.692 |
| | Min-K%++ | 0.672 | 0.676 | 0.652 | 0.649 | 0.674 | 0.680 | 0.696 | 0.697 | **0.722** | 0.724 | 0.691 | 0.702 | 0.686 |
| | DC-PDD | 0.666 | 0.666 | 0.669 | 0.668 | 0.670 | 0.671 | 0.688 | 0.701 | 0.688 | 0.704 | 0.746 | **0.752** | 0.699 |
| | PAC | 0.673 | 0.683 | 0.647 | 0.682 | 0.618 | 0.641 | 0.709 | 0.714 | 0.710 | 0.730 | 0.735 | 0.742 | 0.690 |
| | **AECA** | **0.682** | **0.692** | **0.698** | **0.696** | **0.686** | **0.688** | **0.718** | **0.726** | 0.716 | **0.734** | **0.751** | 0.752 | **0.712** |
| 256 | PPL | 0.666 | 0.637 | 0.674 | 0.628 | 0.639 | 0.596 | 0.692 | 0.658 | 0.688 | 0.653 | 0.713 | 0.697 | 0.662 |
| | Ref | 0.584 | 0.614 | 0.623 | 0.670 | 0.691 | 0.634 | 0.649 | 0.694 | 0.646 | 0.691 | 0.686 | 0.702 | 0.657 |
| | Lowercase | 0.588 | 0.622 | 0.629 | 0.634 | 0.612 | 0.611 | 0.592 | 0.633 | 0.633 | 0.653 | 0.635 | 0.655 | 0.625 |
| | Zlib | 0.679 | 0.526 | 0.686 | 0.518 | 0.662 | 0.505 | 0.703 | 0.553 | 0.701 | 0.548 | 0.729 | 0.575 | 0.615 |
| | Min-K% | 0.701 | 0.684 | 0.706 | 0.692 | 0.669 | 0.633 | 0.710 | 0.700 | 0.721 | 0.683 | 0.743 | 0.719 | 0.697 |
| | Min-K%++ | 0.637 | 0.603 | 0.677 | 0.623 | 0.626 | 0.595 | 0.608 | 0.617 | 0.643 | 0.600 | 0.598 | 0.601 | 0.619 |
| | DC-PDD | 0.626 | 0.618 | 0.670 | 0.658 | 0.625 | 0.598 | 0.664 | 0.682 | 0.687 | 0.677 | 0.737 | 0.680 | 0.660 |
| | PAC | 0.689 | 0.705 | 0.677 | 0.679 | 0.658 | 0.646 | 0.711 | **0.728** | 0.716 | 0.691 | 0.724 | 0.709 | 0.694 |
| | **AECA** | **0.707** | **0.710** | **0.715** | **0.708** | **0.696** | **0.662** | **0.713** | 0.727 | **0.729** | 0.698 | **0.758** | **0.748** | **0.714** |

2.7B(Black et al., 2021), Pythia-2.8B(Biderman et al., 2023), OPT-6.7B(Zhang et al., 2022), Pythia-6.9B(Biderman et al., 2023), Pythia-12B(Biderman et al., 2023), LLaMA-13B(Touvron et al., 2023) and GPT-NeoX-20B(Black et al., 2022). MIMIR is suitable for evaluating models trained on the Pile. Following the benchmark settings of (Duan et al., 2024; Zhang et al., 2025), we conduct experiments using five key parameter scales from the Pythia series: 160M, 1.4B, 2.8B, 6.9B, and 12B.

**Evaluation metrics.** Following prevalent existing studies (Carlini et al., 2021; Duan et al., 2024; Shi et al., 2024; Zhang et al., 2024; 2025), we employ the AUC score (Area Under the ROC Curve) as our evaluation metric. Defined as the area under the receiver operating characteristic curve, AUC comprehensively reflects performance across all possible classification thresholds. Consequently, it provides a holistic and threshold-independent metric that effectively characterizes the method's capability to discriminate between positive and negative samples.

**Implementation details.** To construct the reference corpus $\mathcal{D}_{\text{ref}}$ for calculating token frequency distributions, we utilize a subset of the C4 dataset[2] (Raffel et al., 2020), aligning with the DC-PDD setting. Regarding the hyperparame-

ter configuration, we set $\lambda = 5$ and $\alpha = 1$. As we employ the AUC score as the primary evaluation metric, the determination of a specific threshold $\tau$ is not required for our method. For baseline implementations, following Shi et al. (2024), we set $k = 20$ for Min-K% Prob to achieve optimal performance. Accordingly, to ensure a fair comparison, the hyperparameter $k$ for Min-K%++ is also set to 20. For PAC, the hyperparameters are configured as $k_1 = 5$, $k_2 = 30$, and $m = 0.3 \times |x|$ (where $|x|$ denotes the token count of $x$), consistent with Ye et al. (2024). When a smaller reference model is required, we utilize a correspondingly smaller-scale model, e.g., Pythia-70M for the Pythia series. In particular, Mamba-130M is adopted as the smaller counterpart for Mamba-1.4B. Complete implementation details are provided in Appendix D. The code for AECA is available at: https://github.com/kecy03/AECA.

## 6. Experimental Results

### 6.1. Main Results

**WikiMIA results.** Table 2 presents the main evaluation results based on the AUC metric (Full results are detailed in Appendix H). Experimental results demonstrate that AECA establishes a significant performance advantage on the WikiMIA benchmark. Specifically, regarding average performance across text lengths of 128 and 256, AECA surpasses the leading baselines by approximately 1.3% and

---

[2]The C4 dataset is accessible at: https://huggingface.co/datasets/allenai/c4

*Table 3.* AUC evaluation results on three core subsets (ArXiv, HackerNews, and PubMed Central) of the MIMIR benchmark (Duan et al., 2024) under DeLong's test (Sun & Xu, 2014) ($^*$ indicates $p \leq 0.04$, and $^\dagger$ indicates marginal significance with $0.04 < p \leq 0.05$). Experiments are conducted on a series of Pythia models (Biderman et al., 2023) across varying scales. The proposed AECA method outperforms existing baselines across the majority of experimental settings, achieving significant performance improvements.

| Method | ArXiv | | | | | HackerNews | | | | | PubMed Central | | | | | Avg. |
|---|---|---|---|---|---|---|---|---|---|---|---|---|---|---|---|---|
| | 160M | 1.4B | 2.8B | 6.9B | 12B | 160M | 1.4B | 2.8B | 6.9B | 12B | 160M | 1.4B | 2.8B | 6.9B | 12B | |
| PPL | 0.543 | **0.558** | 0.564 | **0.573** | 0.579 | 0.502 | 0.517 | 0.525 | 0.532 | 0.539 | 0.507 | 0.520 | 0.526 | 0.533 | 0.537 | 0.537$^*$ |
| Ref | 0.494 | 0.532 | 0.539 | 0.553 | 0.563 | 0.488 | 0.518 | 0.535 | **0.546** | **0.558** | 0.498 | 0.524 | 0.530 | 0.537 | 0.541 | 0.530$^*$ |
| Lowercase | 0.529 | 0.543 | 0.552 | 0.558 | 0.566 | 0.494 | 0.508 | 0.517 | 0.525 | 0.523 | 0.523 | 0.526 | 0.527 | 0.531 | 0.534 | 0.530$^*$ |
| Zlib | 0.538 | 0.552 | 0.557 | 0.565 | 0.571 | 0.511 | 0.519 | 0.524 | 0.527 | 0.532 | 0.513 | 0.525 | 0.530 | 0.535 | 0.539 | 0.536$^*$ |
| Min-K% | 0.530 | 0.549 | 0.559 | 0.572 | 0.582 | 0.512 | 0.519 | 0.530 | 0.540 | 0.553 | 0.511 | 0.524 | 0.531 | 0.537 | 0.545 | 0.540$^\dagger$ |
| Min-K%++ | 0.521 | 0.543 | 0.559 | 0.568 | 0.576 | 0.511 | 0.515 | 0.527 | 0.542 | 0.565 | 0.513 | 0.522 | 0.529 | 0.542 | **0.551** | 0.539$^*$ |
| DC-PDD | 0.528 | 0.549 | 0.557 | 0.563 | 0.575 | 0.513 | 0.513 | 0.530 | 0.542 | 0.552 | 0.513 | 0.521 | 0.529 | 0.543 | 0.553 | 0.539$^*$ |
| PAC | 0.527 | 0.549 | 0.562 | 0.572 | 0.582 | 0.485 | 0.492 | 0.513 | 0.519 | 0.532 | 0.510 | 0.526 | 0.535 | **0.546** | 0.551 | 0.533$^\dagger$ |
| AECA | **0.548** | 0.553 | **0.569** | 0.571 | **0.587** | **0.518** | **0.523** | **0.531** | 0.541 | **0.558** | **0.526** | **0.528** | **0.544** | 0.539 | 0.547 | **0.546** |

1.7%, respectively. This advantage is further amplified in challenging scenarios involving long texts and large-scale models (e.g., GPT-NeoX-20B, Len. = 256), where AECA outperforms the best baseline by 1.5% and exceeds the traditional PPL metric by a substantial margin of 4.5%. These gains forcefully validate the unique benefit of AECA in long-text detection: unlike baseline methods that rely on local extrema or static token probabilities, AECA effectively extracts global volatility divergence signals from the entire sequence. Furthermore, AECA's consistent superiority on paraphrased texts and non-Transformer architectures (e.g., Mamba) corroborates its exceptional robustness and generalization capabilities in complex scenarios.

**MIMIR results.** Table 3 presents the comparative AUC results of AECA against baseline methods on the three core subsets: ArXiv, HackerNews, and PubMed Central (Additional results are provided in Appendix F). Despite the challenging nature of MIMIR (Zhang et al., 2025), AECA maintains a significant overall performance advantage. In terms of average performance across all subsets and model scales, AECA outperforms the overall runner-up baseline by 0.6% and the traditional PPL metric by approximately 0.9%. Moreover, on the PubMed Central dataset with Pythia-2.8B, AECA outperforms the runner-up PAC by nearly 1.0%. This compellingly demonstrates that even in extreme scenarios with low discriminability, AECA achieves acute detection capabilities by effectively capturing global volatility divergence features.

### 6.2. Ablation Studies

In this section, we conduct comprehensive ablation studies to validate the individual contributions of AECA's core mechanisms and evaluate its overall robustness. Specifically, we investigate *(1) the necessity of token frequency calibration and entropic convolution, (2) the sensitivity of the volatility divergence coefficient λ, (3) the impact of the reference corpus $\mathcal{D}_{ref}$ across varying scales and domains,*

*Table 4.* Ablation study results of AECA. We report average AUC scores for Pythia-2.8B and Pythia-6.9B on both WikiMIA and MIMIR datasets. 'NLL Volatility' denotes analyzing solely the volatility of the raw loss sequence; '+ Calib. & Div.' represents the volatility divergence analysis incorporating Inherent Self-Information calibration; and '+ Conv.' further integrates entropic convolution, corresponding to the complete AECA method. Detailed results are provided in Table 14 of Appendix H.

| CONFIGURATION | WIKIMIA | | MIMIR | | AVG. |
|---|---|---|---|---|---|
| | 2.8B | 6.9B | 2.8B | 6.9B | |
| NLL VOLATILITY | 0.650 | 0.692 | 0.536 | 0.547 | 0.606 |
| + CALIB. & DIV. | 0.670 | 0.694 | 0.544 | **0.552** | 0.615 |
| + CONV. (AECA) | **0.671** | **0.707** | **0.548** | 0.550 | **0.619** |

*and (4) the influence of the Laplace smoothing coefficient* α.

**Impact of Core Components.** To further validate the efficacy of token frequency calibration, convolution, and volatility divergence analysis within AECA, we conducted ablation studies using Pythia-{2.8B, 6.9B} on the WikiMIA and MIMIR datasets. The comparative configurations are as follows:

- **NLL Volatility**: Utilizing solely the volatility of NLL sequence $\mathcal{L}$ as the detection metric. This represents the baseline performance without introducing any external references or transformations.

- **+ Calib. & Div.**: Based on the raw probabilities $\mathcal{P}$, this configuration incorporates Inherent Self-Information for calibration. It measures the volatility divergence between Entropic Potential $\Phi$ and NLL sequence $\mathcal{L}$.

- **+ Conv. (AECA)**: Based on the '+ Calib. & Div.' setting, this introduces convolution, representing the full instantiation of AECA. It not only utilizes the external reference distribution but also captures fluctuation signals via convolution, ultimately calculating the volatil-

ity divergence between Entropic Difference Spectrum $\mathcal{S}$ and NLL sequence $\mathcal{L}$.

As shown in Table 4, the experimental results exhibit a clear step-wise improvement, verifying the necessity of each module. *(i) Effectiveness of Calibration:* Upon introducing Inherent Self-Information calibration, the method achieves an average gain of 0.9% across all evaluated settings. This validates that mitigating objective token frequency interference is fundamental to accurately exposing overfitting signals. *(ii) Superiority of Convolution:* The integration of entropic convolution provides an additional average improvement of 0.4% over the calibrated baseline. This indicates that convolution effectively complements calibration by capturing the fluctuation patterns between tokens, which simple statistical aggregation fails to utilize.

**Sensitivity Analysis of Hyperparameter $\lambda$.** To systematically evaluate this, we conducted additional ablation experiments on the WikiMIA dataset (Length = 128) using the GPT-Neo-2.7B and Pythia-12B models. Specifically, we assessed the detection AUC by varying $\lambda$ across a broad range: 0, 0.1, and integer values from 1 to 10. The results are presented in Table 5.

*Table 5.* Ablation study on the volatility divergence coefficient $\lambda$.

| Model | $\lambda = 0$ | 0.1 | 1 | 2 | 3 | 4 |
|---|---|---|---|---|---|---|
| GPT-Neo-2.7B | 0.599 | 0.606 | 0.662 | 0.689 | 0.697 | 0.697 |
| Pythia-12B | 0.582 | 0.592 | 0.662 | 0.699 | 0.712 | 0.715 |

| Model | $\lambda = 5$ | 6 | 7 | 8 | 9 | 10 |
|---|---|---|---|---|---|---|
| GPT-Neo-2.7B | **0.698** | **0.698** | 0.692 | 0.690 | 0.688 | 0.687 |
| Pythia-12B | 0.716 | **0.717** | 0.716 | 0.715 | 0.715 | 0.714 |

As shown, while the optimal setting for $\lambda$ varies slightly across different models, it generally remains around 5. Meanwhile, we observe that if $\lambda$ is set too small or approaches zero, the detection performance drops significantly. *This empirically verifies Lemma 4.4, demonstrating that $\lambda$ cannot be smaller than its theoretical lower bound to effectively capture volatility divergence.*

**Impact of Reference Corpus $\mathcal{D}_{\text{ref}}$.** To investigate the robustness of AECA regarding the choice of reference distribution, we evaluate the impact of the reference corpus $\mathcal{D}_{\text{ref}}$ across different scales and domains. Following the experimental setup of DC-PDD (Zhang et al., 2024), we utilized subsets of approximately 1GB and 10GB from both the general-domain C4 corpus and the specialized legal-domain Case-law corpus[3]. Experiments were conducted on the WikiMIA dataset using the Pythia-12B model. As shown

---

[3]The Case-law dataset is accessible at: `https://huggingface.co/datasets/HFforLegal/case-law`

in Table 6, while a larger and general-purpose reference corpus (C4 $\approx$ 10GB) yields the optimal performance, the fluctuations across different settings remain small. Specifically, even when constrained to a smaller, cross-domain corpus (Case-law $\approx$ 1GB), AECA still achieves a highly competitive AUC of 0.705. This demonstrates that AECA is robust and does not require an exhaustive search for the perfect reference distribution to remain effective.

*Table 6.* Ablation study of the reference corpus $\mathcal{D}_{\text{ref}}$.

| $\mathcal{D}_{\text{ref}}$ | C4 | | Case-law | |
|---|---|---|---|---|
| Scale | $\approx$ 1GB | $\approx$ 10GB | $\approx$ 1GB | $\approx$ 10GB |
| Pythia-12B | 0.711 | **0.716** | 0.705 | 0.713 |

**Sensitivity of Smoothing Coefficient $\alpha$.** In addition to the reference corpus, we evaluate the sensitivity of the Laplace smoothing coefficient $\alpha$, which is utilized to address the zero-probability problem during the token frequency estimation process. We vary the value of $\alpha$ across $\{0.01, 0.1, 0.5, 1, 2, 3\}$. Consistent with previous ablation settings, these experiments are conducted on the WikiMIA dataset using the Pythia-12B model. As presented in Table 7, the performance of the AECA method remains stable across a wide range of smoothing factors, with the AUC fluctuating minimally between 0.716 and 0.717. This observation indicates that our framework is highly robust to the choice of $\alpha$. Consequently, the default setting of $\alpha = 1.0$ adopted in our main experiments provides a reliable and stable calibration for detection.

*Table 7.* Ablation study on the smoothing coefficient $\alpha$.

| $\alpha$ | **0.01** | **0.1** | **0.5** | **1** | **2** | **3** |
|---|---|---|---|---|---|---|
| Pythia-12B | 0.716 | 0.717 | 0.717 | 0.716 | 0.716 | 0.716 |

### 6.3. Analysis Study

We further investigate the factors influencing the difficulty of detection, focusing on *target model size* and *text length*, and compare the computational efficiency of various methods.

**Model Size.** We evaluated the average performance of all methods on the three MIMIR subsets using Pythia models of varying scales, specifically 160M, 1.4B, 2.8B, 6.9B, and 12B. As illustrated in Figure 3a, a clear positive correlation is evident: detection AUC scores for all methods improve markedly as model size increases, with AECA consistently maintaining a superior lead across all scales. This trend corroborates findings from prior research (Shi et al., 2024; Zhang et al., 2024; Ye et al., 2024), indicating that larger models possess stronger learning capacities and tend to memorize pretraining data more intensively, thereby leaving

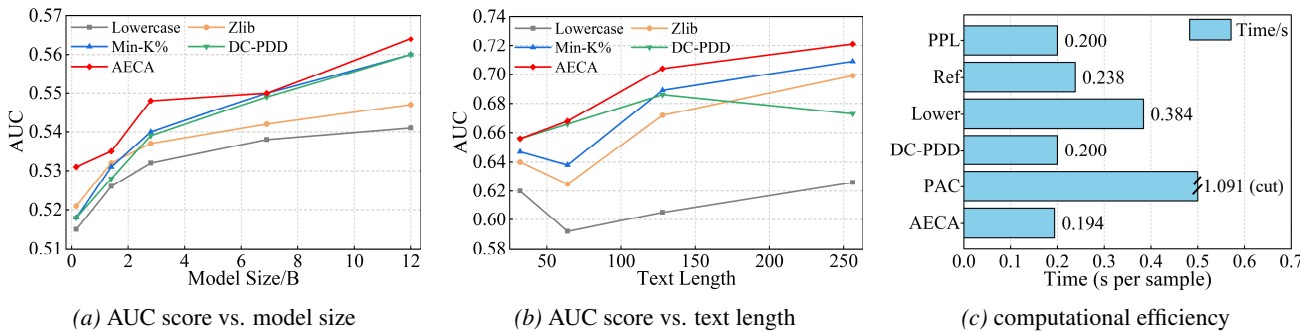

*(a)* AUC score vs. model size      *(b)* AUC score vs. text length      *(c)* computational efficiency

*Figure 3.* **Comprehensive analysis of detection performance and efficiency.** We examine the scalability and robustness of AECA from three perspectives: *(a) detection performance across different LLM model sizes, (b) sensitivity to varying input text lengths, and (c) average inference time per sample measured in seconds.* Across these settings, AECA consistently achieves strong detection performance, exhibits robust behavior under length variation, and maintains competitive computational efficiency.

behind more distinguishable statistical traces. (Full results are detailed in Table 12 of Appendix H.)

**Sample Length.** Furthermore, we evaluated the impact of sample length on detection performance based on the WikiMIA dataset under its *original* setting. As shown in Figure 3b, the AUC of all methods exhibits an upward trend as text length increases. This is primarily because longer texts contain more characteristic information memorized by the model, making them more distinguishable from unseen texts. Notably, while baseline methods typically suffer from significant performance degradation in short-text scenarios, AECA not only performs robustly but also demonstrates continuous performance improvement with increasing length. *This further corroborates the advantage of AECA in effectively enhancing detection capabilities through a global sequence perspective.*

**Computational Efficiency.** Additionally, we recorded the average inference time (in seconds) for detecting a single sample for each method under identical conditions, as illustrated in Figure 3c. Existing high-performance methods (e.g., PAC) require generating multiple augmented samples to calculate polarization distance or rely on additional reference models (e.g., Ref), thereby introducing substantial computational overhead. In contrast, AECA achieves state-of-the-art detection performance while incurring only minimal computational costs. Its inference speed is comparable to the simple Perplexity (PPL) baseline, rendering AECA highly practical for large-scale, real-time pretraining data detection tasks.

## 7. Conclusion

In this paper, we propose AECA, a method for detecting pretraining data in LLMs without accessing the training corpus. Specifically, to address the limitations of prior methods that rely on local statistics of specific token sub-

sets—overlooking the probability dynamics during the token generation process of the entire text—we adopt a global sequence perspective. We model probability sequences as dynamic signals to capture their global volatility characteristics. Furthermore, we design a framework integrating calibration with convolutional filtering. Grounded in the theory of Volatility Divergence, this approach effectively distinguishes between memorization and reasoning-based generation, supported by theoretical proofs. Extensive experiments demonstrate that AECA combines both accuracy and robustness. It effectively detects pretraining data and exhibits superior performance compared to baselines, particularly on long-text scenarios, laying a solid foundation for future research.

## Acknowledgments

This work was supported by grants from the National Key Research and Development Program of China (Grant No. 2024YFC3308200), the National Natural Science Foundation of China (U25B2072), the Key Technologies R & D Program of Anhui Province (No. 202423k09020039) and the Fundamental Research Funds for the Central Universities.

## Impact Statement

The core contribution of this work lies in proposing a novel LLM pretraining data detection paradigm that shifts focus from isolated token statistics to dynamics in the generation process. While we do not anticipate direct negative societal impacts, the potential for false positives or negatives means our scores should not be directly treated as definitive legal or factual evidence of data exposure. Looking ahead, we aim to broaden the scope of detection granularity: extending our framework from sample-level verification to corpus-level detection, determining whether a model has been exposed to entire datasets during pretraining.

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

# A. Proof of Proposition 4.3

**Proposition A.1.** *The Entropic Difference Response acts as a high-pass operator sensitive to abrupt variations in Entropic Potential. It theoretically isolates memorization traces by suppressing the smooth, low-frequency components characteristic of natural semantic generation, while selectively amplifying the sharp, high-frequency oscillations induced by overfitting.*

*Proof.* Following standard digital signal processing principles (Oppenheim, 1999), we begin by modeling the Entropic Potential sequence $\Phi$ as a discrete-time signal $x[n]$ and the Entropic Difference Response operation as a linear time-invariant (LTI) system. According to the definition of the difference response (Definition 4.2), this operation corresponds to a first-order difference operator. While the forward difference is used in implementation, in the context of spectral magnitude analysis, this is symmetric to a backward difference. Thus, the impulse response is implicitly given by:

$$h[n] = \delta[n] - \delta[n-1], \tag{8}$$

where $\delta[\cdot]$ is the Kronecker delta function.

First, to analyze the spectral properties, we derive the Z-transform of the kernel $h[n]$:

$$H(z) = \sum_{n=-\infty}^{\infty} h[n]z^{-n} = 1 - z^{-1}. \tag{9}$$

Next, we evaluate the frequency response by substituting $z = e^{j\omega}$, where $\omega \in [0, \pi]$ represents the normalized angular frequency. By factoring out the half-angle term $e^{-j\omega/2}$, we obtain:

$$\begin{aligned} H(e^{j\omega}) &= 1 - e^{-j\omega} \\ &= e^{-j\omega/2}\left(e^{j\omega/2} - e^{-j\omega/2}\right). \end{aligned} \tag{10}$$

Based on Euler's formula, we have: $e^{j\omega/2} = \cos(\frac{\omega}{2}) + j\sin(\frac{\omega}{2})$, $e^{-j\omega/2} = \cos(\frac{\omega}{2}) - j\sin(\frac{\omega}{2})$. Thus:

$$H(e^{j\omega}) = e^{-j\omega/2}\left(2j\sin\left(\frac{\omega}{2}\right)\right). \tag{11}$$

Subsequently, we derive the Magnitude Response $|H(e^{j\omega})|$ by taking the modulus of each factor. Utilizing the properties that the modulus of a complex exponential on the unit circle is 1 (i.e., $|e^{j\theta}| = 1$) and the modulus of the imaginary unit is $|2j| = 2$, we have:

$$\begin{aligned} |H(e^{j\omega})| &= \left|e^{-j\omega/2}\right| \cdot |2j| \cdot \left|\sin\left(\frac{\omega}{2}\right)\right| \\ &= 1 \cdot 2 \cdot \left|\sin\left(\frac{\omega}{2}\right)\right| \\ &= 2\left|\sin\left(\frac{\omega}{2}\right)\right|. \end{aligned} \tag{12}$$

Finally, we analyze the filter's behavior at critical frequency limits:

- **Low-Frequency Limit ($\omega \to 0$):** This corresponds to smooth, stable text generation (characteristic of expected generalization).
$$\lim_{\omega \to 0} |H(e^{j\omega})| = 2\sin(0) = 0. \tag{13}$$
The gain is zero, indicating that the filter completely suppresses constant or slowly varying semantic signals (DC bias).

- **High-Frequency Limit ($\omega \to \pi$):** This corresponds to drastic token-to-token fluctuations (characteristic of memorization anomalies).
$$\lim_{\omega \to \pi} |H(e^{j\omega})| = 2\sin(\pi/2) = 2. \tag{14}$$
The gain is maximized at the Nyquist frequency, meaning the filter selectively amplifies the rapid singularities associated with memorization anomalies.

**Conclusion:** The derived magnitude profile confirms that the system acts as an ideal high-pass filter, mathematically validating the design of the Entropic Difference Response in capturing volatility divergence. $\square$

# B. Proofs of Lemma 4.4

**Lemma B.1** (Volatility Divergence). *Assume that for a valid text sequence $x$ in the natural language domain, the volatility of its self-information possesses a lower bound $\mu$ (i.e., $\sigma(\mathcal{I}_x) \geq \mu$). Let the Generalization State be $\mathbb{G}$ with fitting error bound $\gamma$, and the Memorization State be $\mathbb{M}$ with an empirical error bound $\epsilon \ll \gamma$. Then, there exists a coefficient $\lambda > \frac{(2\gamma+\epsilon) \log \frac{N+\alpha|\mathcal{V}|}{\alpha} + \log \frac{N+\alpha}{\alpha}}{\mu - (\gamma + \frac{\epsilon}{2})}$ such that:*

$$[\sigma(\mathcal{S}_x) - \lambda \sigma(\mathcal{L}_x)]_{\mathbb{M}} > [\sigma(\mathcal{S}_x) - \lambda \sigma(\mathcal{L}_x)]_{\mathbb{G}}. \tag{15}$$

*Proof.* To rigorously prove this lemma, we define inherent self-information as $\mathcal{I}_t \triangleq \mathcal{I}_{\text{self}}(x_t)$. The detection metric is defined as $\mathbf{S} = \sigma(\mathcal{S}) - \lambda \cdot \sigma(\mathcal{L})$, where $\mathcal{S}$ denotes the Entropic Difference Spectrum and $\mathcal{L}$ denotes the Negative Log-Likelihood (NLL) sequence.

**Preliminaries: Boundedness of Self-Information**

Based on Eq.(3), since the reference corpus $\mathcal{D}_{\text{ref}}$ is finite in size and Laplace Smoothing is applied, the reference probability $P_{\text{ref}}(x_t)$ is bounded for any $x_t$:

$$\frac{\alpha}{N + \alpha|\mathcal{V}|} \leq P_{\text{ref}}(x_t) = \frac{C(x_t; \mathcal{D}_{\text{ref}}) + \alpha}{N + \alpha|\mathcal{V}|} \leq \frac{N + \alpha}{N + \alpha|\mathcal{V}|}. \tag{16}$$

Consequently, the self-information $\mathcal{I}_t = -\log P_{\text{ref}}(x_t)$ is strictly bounded:

$$\mathcal{I}_t \leq -\log \frac{\alpha}{N + \alpha|\mathcal{V}|} = \log \frac{N + \alpha|\mathcal{V}|}{\alpha} \triangleq \mathcal{I}_{\max}, \tag{17}$$

$$\mathcal{I}_t \geq -\log \frac{N + \alpha}{N + \alpha|\mathcal{V}|} = \log \frac{N + \alpha|\mathcal{V}|}{N + \alpha} \triangleq \mathcal{I}_{\min}. \tag{18}$$

The first-order difference $\Delta\mathcal{I}_t = \mathcal{I}_t - \mathcal{I}_{t+1}$ is thus confined within the interval:

$$\Delta\mathcal{I}_t \in [\mathcal{I}_{\min} - \mathcal{I}_{\max}, \ \mathcal{I}_{\max} - \mathcal{I}_{\min}]. \tag{19}$$

Let the total length of the interval be $R_\Delta = (\mathcal{I}_{\max} - \mathcal{I}_{\min}) - (\mathcal{I}_{\min} - \mathcal{I}_{\max}) = 2(\mathcal{I}_{\max} - \mathcal{I}_{\min})$. According to Popoviciu's Inequality (Popoviciu, 1935), the standard deviation of the inherent volatility is bounded by:

$$\sigma(\Delta\mathcal{I}) \leq \frac{R_\Delta}{2} = \mathcal{I}_{\max} - \mathcal{I}_{\min} = \log \frac{N + \alpha|\mathcal{V}|}{\alpha} - \log \frac{N + \alpha|\mathcal{V}|}{N + \alpha} = \log \frac{N + \alpha}{\alpha}. \tag{20}$$

Similarly, for self-information $\mathcal{I}_t$:

$$\sigma(\mathcal{I}) \leq \frac{\mathcal{I}_{\max} - \mathcal{I}_{\min}}{2} = \frac{1}{2}\left(\log \frac{N + \alpha|\mathcal{V}|}{\alpha} - \log \frac{N + \alpha|\mathcal{V}|}{N + \alpha}\right) = \frac{1}{2}\log \frac{N + \alpha}{\alpha}. \tag{21}$$

**Case 1: Memorization State ($\mathbb{M}$)**

Under the memory mechanism, due to overfitting, the model assigns high probabilities to memorized tokens (Carlini et al., 2021). We constrain the prediction error by a small empirical upper bound $\epsilon$ ($\epsilon \ll \gamma$):

$$0 \leq \mathcal{L}_t = -\log p_t \leq \epsilon. \tag{22}$$

According to Popoviciu's Inequality, the standard deviation of the NLL sequence is strictly bounded by half its range:

$$\sigma(\mathcal{L})_{\mathbb{M}} \leq \frac{\epsilon - 0}{2} = \frac{\epsilon}{2}. \tag{23}$$

Next, we analyze the entropy difference spectrum $\mathcal{S}_t = p_t \mathcal{I}_t - p_{t+1} \mathcal{I}_{t+1} = e^{-\mathcal{L}_t} \mathcal{I}_t - e^{-\mathcal{L}_{t+1}} \mathcal{I}_{t+1}$. Since $0 \leq \mathcal{L}_t \leq \epsilon$, we have $e^{-\mathcal{L}_t} = 1 - \delta_t$, where $0 \leq \delta_t \leq \epsilon$. Substituting this into $\mathcal{S}_t$:

$$\mathcal{S}_t = (1 - \delta_t)\mathcal{I}_t - (1 - \delta_{t+1})\mathcal{I}_{t+1} = \Delta\mathcal{I}_t - (\delta_t \mathcal{I}_t - \delta_{t+1}\mathcal{I}_{t+1}). \tag{24}$$

Let the error term be $\xi_t = \delta_t \mathcal{I}_t - \delta_{t+1} \mathcal{I}_{t+1}$. Since $0 \leq \delta_t \leq \epsilon$ and $\mathcal{I}_t \leq \mathcal{I}_{\max}$, each term $\delta_t \mathcal{I}_t \in [0, \epsilon \mathcal{I}_{\max}]$, making the range of $\xi_t$ bounded within $[-\epsilon \mathcal{I}_{\max}, \epsilon \mathcal{I}_{\max}]$. Using Popoviciu's Inequality again, its standard deviation satisfies:

$$\sigma(\xi) \leq \frac{2\epsilon \mathcal{I}_{\max}}{2} = \epsilon \mathcal{I}_{\max}. \tag{25}$$

Applying the reverse triangle inequality (i.e., $\sigma(X + Y) \geq \sigma(X) - \sigma(Y)$), we establish the lower bound for the volatility of the entropy spectrum:

$$\sigma(\mathcal{S})_{\mathbb{M}} = \sigma(\Delta \mathcal{I} - \xi) \geq \sigma(\Delta \mathcal{I}) - \sigma(\xi) \geq \sigma(\Delta \mathcal{I}) - \epsilon \mathcal{I}_{\max}. \tag{26}$$

Thus, the lower bound of the score in the memory state is:

$$\mathbf{S}_{\mathbb{M}} = \sigma(\mathcal{S})_{\mathbb{M}} - \lambda \cdot \sigma(\mathcal{L})_{\mathbb{M}} \geq (\sigma(\Delta \mathcal{I}) - \epsilon \mathcal{I}_{\max}) - \lambda \frac{\epsilon}{2}. \tag{27}$$

**Case 2: Generalization State ($\mathbb{G}$)**

In the generalization state, the model has learned the statistical regularities of the language, predicting probabilities with bounded error. For any $t$, $\mathcal{L}_t > 0$, $\mathcal{I}_t > 0$, and $|\mathcal{L}_t - \mathcal{I}_t| \leq \gamma$.

Using the triangle inequality to decompose the entropy difference term $\mathcal{S}_t$, we introduce intermediate terms $\mathcal{I}_t e^{-\mathcal{I}_t}$ and $\mathcal{I}_{t+1} e^{-\mathcal{I}_{t+1}}$ for rearrangement:

$$
\begin{aligned}
|\mathcal{S}_t| &= |\mathcal{I}_t p_t - \mathcal{I}_{t+1} p_{t+1}| \\
&= |\mathcal{I}_t e^{-\mathcal{L}_t} - \mathcal{I}_{t+1} e^{-\mathcal{L}_{t+1}}| \\
&= |(\mathcal{I}_t e^{-\mathcal{L}_t} - \mathcal{I}_t e^{-\mathcal{I}_t}) + (\mathcal{I}_t e^{-\mathcal{I}_t} - \mathcal{I}_{t+1} e^{-\mathcal{I}_{t+1}}) + (\mathcal{I}_{t+1} e^{-\mathcal{I}_{t+1}} - \mathcal{I}_{t+1} e^{-\mathcal{L}_{t+1}})| \\
&\leq \underbrace{|\mathcal{I}_t e^{-\mathcal{L}_t} - \mathcal{I}_t e^{-\mathcal{I}_t}|}_{(1)} + \underbrace{|\mathcal{I}_t e^{-\mathcal{I}_t} - \mathcal{I}_{t+1} e^{-\mathcal{I}_{t+1}}|}_{(2)} + \underbrace{|\mathcal{I}_{t+1} e^{-\mathcal{I}_{t+1}} - \mathcal{I}_{t+1} e^{-\mathcal{L}_{t+1}}|}_{(3)}.
\end{aligned}
\tag{28}
$$

*Analysis of Terms (1) and (3):* Consider the function $g(x) = e^{-x}$. Its derivative satisfies $|g'(x)| \leq 1$ for $x \geq 0$; thus, it is a Lipschitz continuous function with constant $K = 1$. Hence, $|e^{-\mathcal{L}_t} - e^{-\mathcal{I}_t}| \leq K \cdot |\mathcal{L}_t - \mathcal{I}_t| \leq \gamma$. Multiplying by the bounded $\mathcal{I}_t \leq \mathcal{I}_{\max}$, we obtain:

$$|\mathcal{I}_t e^{-\mathcal{L}_t} - \mathcal{I}_t e^{-\mathcal{I}_t}| = \mathcal{I}_t |e^{-\mathcal{L}_t} - e^{-\mathcal{I}_t}| \leq \mathcal{I}_{\max} \cdot \gamma, \tag{29}$$

$$|\mathcal{I}_{t+1} e^{-\mathcal{I}_{t+1}} - \mathcal{I}_{t+1} e^{-\mathcal{L}_{t+1}}| = \mathcal{I}_{t+1} |e^{-\mathcal{L}_{t+1}} - e^{-\mathcal{I}_{t+1}}| \leq \mathcal{I}_{\max} \cdot \gamma. \tag{30}$$

*Analysis of Term (2):* Consider the function $h(x) = xe^{-x}$. Its derivative satisfies $|h'(x)| = |(1 - x)e^{-x}| \leq 1$ for $x \geq 0$; thus, it is Lipschitz continuous with constant $K' = 1$. Hence:

$$|\mathcal{I}_t e^{-\mathcal{I}_t} - \mathcal{I}_{t+1} e^{-\mathcal{I}_{t+1}}| = |h(\mathcal{I}_t) - h(\mathcal{I}_{t+1})| \leq K' \cdot |\mathcal{I}_t - \mathcal{I}_{t+1}| = |\Delta \mathcal{I}_t|. \tag{31}$$

From Eq.(19), we know that $|\Delta \mathcal{I}_t| \leq \log \frac{N+\alpha}{\alpha}$. Combining Eq.(28, 29, 30, 31), the magnitude of $\mathcal{S}_t$ is strictly bounded by:

$$|\mathcal{S}_t| \leq 2\gamma \mathcal{I}_{\max} + \log \frac{N + \alpha}{\alpha}. \tag{32}$$

This inequality implies that $\mathcal{S}_t \in \left[ -(2\gamma \mathcal{I}_{\max} + \log \frac{N+\alpha}{\alpha}), \ 2\gamma \mathcal{I}_{\max} + \log \frac{N+\alpha}{\alpha} \right]$. According to Popoviciu's Inequality, the standard deviation of $\mathcal{S}_t$ is bounded by its half-range:

$$\sigma(\mathcal{S})_{\mathbb{G}} \leq 2\gamma \mathcal{I}_{\max} + \log \frac{N + \alpha}{\alpha}. \tag{33}$$

Let $\mathcal{L}_t = \mathcal{I}_t + \epsilon_t$, where $|\epsilon_t| \leq \gamma$. Since the error term is bounded within $[-\gamma, \gamma]$, Popoviciu's inequality implies $\sigma(\epsilon) \leq \gamma$. Combined with the reverse triangle inequality (i.e., $\sigma(X + Y) \geq \sigma(X) - \sigma(Y)$), we obtain the lower bound for loss volatility:

$$\sigma(\mathcal{L})_{\mathbb{G}} \geq \sigma(\mathcal{I}) - \sigma(\epsilon) \geq \sigma(\mathcal{I}) - \gamma. \tag{34}$$

Therefore, the upper bound of the score in the generalization state is:

$$\mathbf{S}_{\mathbb{G}} = \sigma(\mathcal{S})_{\mathbb{G}} - \lambda\sigma(\mathcal{L})_{\mathbb{G}} \leq 2\gamma\mathcal{I}_{\max} + \log\frac{N+\alpha}{\alpha} - \lambda(\sigma(\mathcal{I}) - \gamma). \tag{35}$$

**Conclusion**

To distinguish memorized data from unseen generalized data, we require $\mathbf{S}_{\mathbb{M}} > \mathbf{S}_{\mathbb{G}}$. Combining Eq.(27) and Eq.(35), we have:

$$(\sigma(\Delta\mathcal{I}) - \epsilon\mathcal{I}_{\max}) - \lambda\frac{\epsilon}{2} > \left(2\gamma\mathcal{I}_{\max} + \log\frac{N+\alpha}{\alpha}\right) - \lambda(\sigma(\mathcal{I}) - \gamma). \tag{36}$$

Rearranging terms to isolate $\lambda$:

$$\lambda\left(\sigma(\mathcal{I}) - \gamma - \frac{\epsilon}{2}\right) > (2\gamma + \epsilon)\mathcal{I}_{\max} + \log\frac{N+\alpha}{\alpha} - \sigma(\Delta\mathcal{I}). \tag{37}$$

Considering an effective generalization state where the model's fitting error is lower than the inherent self-information volatility, and given that $\epsilon \ll \gamma$, it follows that $\sigma(\mathcal{I}) \geq \mu > \gamma + \frac{\epsilon}{2}$. Solving for $\lambda$:

$$\lambda > \frac{(2\gamma + \epsilon)\mathcal{I}_{\max} + \log\frac{N+\alpha}{\alpha} - \sigma(\Delta\mathcal{I})}{\sigma(\mathcal{I}) - \left(\gamma + \frac{\epsilon}{2}\right)}. \tag{38}$$

By utilizing the non-negativity condition $\sigma(\Delta\mathcal{I}) \geq 0$ and substituting the expression for $\mathcal{I}_{\max}$ (Eq.(17)) along with the lower bound of $\sigma(\mathcal{I})$, we obtain:

$$\lambda > \frac{(2\gamma + \epsilon)\log\frac{N+\alpha|\mathcal{V}|}{\alpha} + \log\frac{N+\alpha}{\alpha}}{\mu - \left(\gamma + \frac{\epsilon}{2}\right)}. \tag{39}$$

Thus, there exists a $\lambda$ satisfying $\mathbf{S}_{\mathbb{M}} > \mathbf{S}_{\mathbb{G}}$. This completes the proof. $\qquad\square$

# C. Details of Benchmarks

In this section, we provide a detailed introduction to the benchmark datasets utilized in this paper: WikiMIA (Shi et al., 2024) and MIMIR (Duan et al., 2024). Specific statistics are presented in Table 1.

### C.1. WikiMIA Benchmark.

Constructed from Wikipedia, this dataset utilizes a temporal split to distinguish between training and non-training data. Specifically, training texts are selected from Wikipedia articles published prior to 2017, while non-training texts are derived from events occurring after 2023. The training cutoff dates of the models examined in this paper fall strictly within this interval, ensuring they have not been exposed to the non-training texts (Zhang et al., 2025). The WikiMIA benchmark comprises text data of four distinct lengths: 32, 64, 128, and 256 words; this fine-grained segmentation allows us to evaluate method performance across varying context lengths. The detailed distribution is illustrated in Figure 4. Furthermore, WikiMIA includes a paraphrased version to assess whether detection methods can effectively identify semantically equivalent samples, thereby simulating the detection of modified contamination. However, as the official paraphrased version is not yet publicly available, we employed ChatGPT for the rewriting process in our experiments. The prompt used for sentence rewriting is as follows:

> " Rewrite the following text with the smallest possible changes to wording and structure—keep the semantics identical, do not add/delete any information, maintain a similar length (word count difference $\leq 10\%$), and ensure the BLEU score between the original and rewritten text is greater than 0.8. Output only the rewritten text, no extra explanation. Original text: {text}
>
> Rewritten text: "

### C.2. MIMIR Benchmark.

Constructed upon the Pile, this dataset defines training and non-training texts directly based on the official training and test splits provided by the Pile. The Pythia models examined here strictly adhere to the evaluation settings of the MIMIR

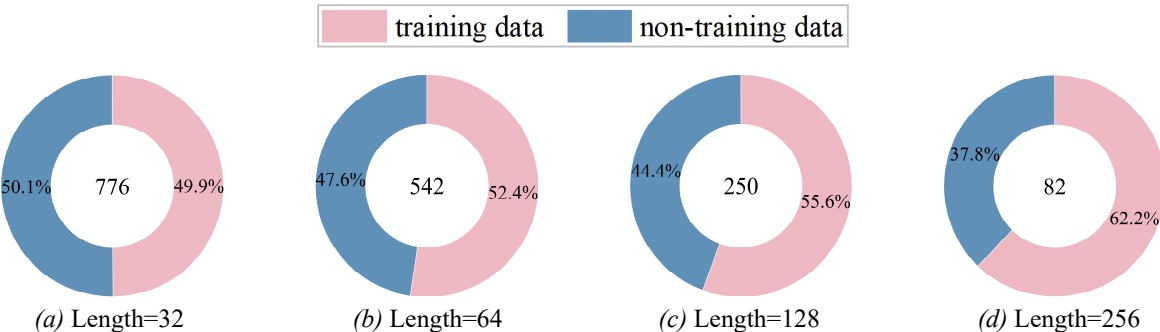

*Figure 4.* Distribution of training and non-training samples in the WikiMIA dataset across different lengths.

study (Duan et al., 2024), having been trained exclusively on the Pile's training set without exposure to the test set data. In our experiments, we selected three representative subsets—*ArXiv, HackerNews, and PubMed Central*—and applied a strict 13-gram overlap threshold as the filtering criterion; specifically, the 13-gram overlap rate between non-member and member samples is controlled below 20%. This configuration effectively mitigates distributional bias arising from high text overlap while preserving the difficulty of privacy evaluation in realistic scenarios, thereby enabling a more precise assessment of the effectiveness of detection methods. These subsets cover a broad spectrum of linguistic styles:

- **ArXiv**: Represents highly technical and formal academic texts typically containing an abundance of complex mathematical symbols and domain-specific terminology.

- **PubMed Central**: Comprises biomedical literature and clinical trial reports, representing specialized scientific discourse distinct from general web text.

- **HackerNews**: Contains user-generated conversational texts from forum discussions. This domain is characterized by informal language, posing unique challenges for detection methods.

## D. Experimental Implementation Details.

In this section, we detail the experimental setup. Following the setup in DC-PDD (Zhang et al., 2024), we constructed the reference corpus $\mathcal{D}_{\text{ref}}$ using the *realnewslike* subset ($\approx$15GB) of the C4 dataset to estimate token frequency distributions. Regarding the hyperparameter configuration, we set $\alpha = 1$. For $\lambda$, we conducted a grid search over the integer range $[1, 10]$ and selected $\lambda = 5$ for optimal performance. As we employ the AUC score as the primary evaluation metric, the determination of a specific threshold $\tau$ is not required for our method. For settings requiring a reference model, following Zhang et al. (2025), we employed corresponding smaller-scale models: Mamba-130M for Mamba-1.4B, OPT-350M for OPT-6.7B, Pythia-70M for the Pythia series, LLaMA-7B for LLaMA-13B, and GPT-Neo-125M for both GPT-Neo-2.7B and GPT-NeoX-20B.

## E. Limitations

In this section, we discuss the limitations of our study. Although AECA demonstrates promising results in detecting pretraining data for LLMs, several constraints remain, which we plan to address in future work.

- **Hyperparameter Sensitivity.** A critical component of our Volatility Divergence theory is the coefficient $\lambda$, which serves to calibrate the scale disparity between the Entropic Difference Spectrum and the Negative Log-Likelihood (NLL) sequence. While Lemma 4.4 establishes a theoretical lower bound for its existence, the expression reveals that its specific value is influenced by the model's generalization capability and the reference corpus. Currently, we rely on empirical search to determine this parameter. Future work could explore adaptive mechanisms to dynamically estimate $\lambda$ based on specific characteristics (Liu et al., 2024).

- **Access Requirements.** Our method relies on access to the model's output probability distributions. Consequently, this approach faces challenges with closed-source models that typically provide only restricted access to partial token

probabilities (e.g., top-logprobs) via APIs, which remains a common constraint shared by current gray-box detection paradigms. With an increasing number of commercial LLMs restricting probability access, detecting data contamination becomes more difficult. Therefore, we advocate for model providers to offer greater transparency regarding output probabilities or embrace open weights, thereby fostering the sustainable development of trustworthy AI.

- **Scale and Verification Constraints.** Finally, due to computational resource limitations, our experimental validation was primarily conducted on models with up to 20B parameters. While AECA exhibits consistent performance gains at this scale, validating its efficacy on larger foundation models (e.g., 70B+ parameters) is crucial. Future work will focus on scaling experiments and collaborating with industry partners to validate AECA on larger architectures.

- **Detection Constraints on Short Texts.** While AECA excels in long-text scenarios, its efficacy on shorter texts (e.g., Length = 32) is limited. Because AECA relies on sequence-level volatility, statistically insufficient sampling in short texts causes local word frequency noise to overwhelm global fluctuation patterns, reducing the discriminability of $\sigma(\mathcal{S})$. In contrast, local methods remain highly sensitive in these scenarios by isolating specific anomalous tokens. Future work will explore hybrid mechanisms integrating AECA's global stability with token-level local sensitivity.

## F. Additional Results on MIMIR

To further assess whether the improvements hold consistently across diverse conditions and to address concerns regarding temporal, lexical, and structural biases, we report additional evaluation results on the remaining four subsets of the MIMIR benchmark: Wikipedia, Pile CC, Github, and DM Mathematics. These subsets cover a broader range of domains, including standard text as well as low-entropy technical data (code and mathematics). As shown in Table 8, AECA maintains an overall performance advantage across the full range of evaluated settings, demonstrating the robustness of our method.

*Table 8.* AUC evaluation results on the remaining subsets (Wikipedia, Pile CC, Github, and DM Mathematics) of the MIMIR benchmark.

| Method | Wikipedia | | | | | Pile CC | | | | | Github | | | | | DM Mathematics | | | | |
|---|---|---|---|---|---|---|---|---|---|---|---|---|---|---|---|---|---|---|---|---|
| | 160M | 1.4B | 2.8B | 6.9B | 12B | 160M | 1.4B | 2.8B | 6.9B | 12B | 160M | 1.4B | 2.8B | 6.9B | 12B | 160M | 1.4B | 2.8B | 6.9B | 12B |
| PPL | 0.511 | 0.535 | 0.543 | 0.560 | 0.571 | 0.502 | 0.512 | 0.514 | 0.523 | 0.529 | 0.641 | 0.696 | 0.710 | 0.721 | 0.732 | 0.527 | 0.533 | 0.535 | 0.535 | 0.544 |
| Ref | 0.499 | 0.531 | 0.540 | 0.564 | 0.575 | 0.500 | 0.524 | 0.525 | 0.532 | 0.540 | 0.619 | 0.613 | 0.626 | 0.632 | 0.641 | 0.462 | 0.475 | 0.481 | 0.472 | 0.465 |
| Lowercase | 0.505 | 0.541 | 0.547 | 0.567 | 0.579 | 0.514 | 0.520 | 0.529 | **0.538** | 0.545 | 0.688 | 0.712 | 0.724 | 0.735 | 0.742 | **0.558** | 0.565 | 0.567 | 0.569 | 0.573 |
| Zlib | 0.502 | 0.533 | 0.541 | 0.561 | 0.572 | 0.511 | 0.523 | 0.524 | 0.532 | 0.537 | 0.685 | **0.714** | 0.735 | 0.744 | 0.755 | 0.535 | 0.550 | 0.552 | 0.552 | 0.561 |
| Min-K% | 0.500 | 0.534 | 0.546 | 0.563 | 0.574 | 0.508 | 0.515 | 0.515 | 0.525 | 0.527 | 0.656 | 0.693 | 0.726 | 0.739 | 0.750 | 0.535 | 0.551 | 0.551 | 0.550 | 0.551 |
| Min-K%++ | 0.510 | 0.548 | 0.559 | 0.580 | **0.603** | 0.512 | 0.518 | 0.521 | 0.535 | 0.542 | 0.652 | 0.689 | 0.687 | 0.728 | 0.736 | 0.552 | 0.565 | **0.578** | 0.579 | 0.573 |
| DC-PDD | 0.502 | 0.539 | 0.548 | 0.569 | 0.578 | 0.514 | 0.516 | 0.522 | 0.520 | 0.534 | 0.687 | 0.705 | 0.735 | 0.734 | 0.760 | 0.546 | 0.542 | 0.554 | 0.556 | 0.561 |
| PAC | 0.519 | 0.542 | 0.556 | 0.574 | 0.579 | **0.519** | 0.524 | 0.528 | 0.528 | 0.547 | 0.659 | 0.682 | 0.706 | 0.724 | 0.729 | 0.549 | 0.554 | 0.558 | 0.558 | 0.577 |
| AECA | **0.520** | **0.554** | **0.563** | **0.583** | 0.597 | 0.518 | **0.530** | **0.533** | 0.535 | **0.558** | **0.699** | 0.711 | **0.744** | **0.756** | **0.769** | 0.558 | **0.572** | 0.574 | **0.582** | **0.584** |

## G. Empirical Validation of Volatility Divergence

To intuitively illustrate and empirically validate the volatility-divergence hypothesis—which was theoretically established in Lemma 4.4—we analyze the distributions of the Entropic Difference Spectrum volatility $\sigma(\mathcal{S})$ and the Negative Log-Likelihood (NLL) volatility $\sigma(\mathcal{L})$ for both member and non-member samples. As an illustrative example, Table 9 presents the average $\sigma(\mathcal{S})$ and $\sigma(\mathcal{L})$ calculated using the Pythia-12B model on the challenging ArXiv dataset.

*Table 9.* Average volatilities on the ArXiv dataset.

| Metric | Memorized | Non-Memorized |
|---|---|---|
| $\sigma(\mathcal{S})$ | 4.021 | 3.849 |
| $\sigma(\mathcal{L})$ | 1.798 | 2.035 |
| $\Delta$ | 2.223 | 1.814 |

As demonstrated in the table, for non-memorized samples, these two volatilities are coupled. Conversely, for memorized samples, this coupling is broken: the loss volatility $\sigma(\mathcal{L})$ collapses to a lower value, whereas the entropic difference volatility $\sigma(\mathcal{S})$ remains highly volatile, resulting in a larger divergence margin $\Delta$. This observation aligns with our theoretical derivations. As explained in Section 4.2, the rationale behind why this coupling breaks for memorized samples is that the model anomalously assigns high probabilities even to inherently difficult-to-predict tokens (i.e., tokens with high self-information) within memorized texts. This anomalous memorization phenomenon serves as the primary statistical signal leveraged by the AECA framework.

## H. Detailed Results

*Table 10.* Full AUC results on the WikiMIA benchmark under the *original* setting.

**(a) Length = 32**

| Model | PPL | Ref | Lowercase | Zlib | Min-K% | Min-K%++ | DC-PDD | PAC | AECA |
|---|---|---|---|---|---|---|---|---|---|
| Mamba-1.4B | 0.610 | 0.584 | 0.603 | 0.620 | 0.619 | 0.649 | 0.650 | 0.665 | 0.648 |
| GPT-Neo-2.7B | 0.613 | 0.597 | 0.599 | 0.620 | 0.630 | 0.673 | 0.643 | 0.662 | 0.644 |
| Pythia-2.8B | 0.616 | 0.630 | 0.605 | 0.625 | 0.625 | 0.635 | 0.648 | 0.668 | 0.637 |
| OPT-6.7B | 0.604 | 0.666 | 0.603 | 0.611 | 0.614 | 0.651 | 0.648 | 0.657 | 0.635 |
| Pythia-6.9B | 0.645 | 0.655 | 0.628 | 0.650 | 0.682 | 0.715 | 0.684 | 0.710 | 0.689 |
| Pythia-12B | 0.656 | 0.662 | 0.655 | 0.659 | 0.678 | 0.713 | 0.677 | 0.715 | 0.680 |
| LLaMA-13B | 0.657 | 0.638 | 0.595 | 0.660 | 0.642 | 0.803 | 0.596 | 0.698 | 0.627 |
| GPT-NeoX-20B | 0.674 | 0.607 | 0.670 | 0.678 | 0.687 | 0.698 | 0.705 | 0.745 | 0.688 |
| Average | 0.634 | 0.630 | 0.620 | 0.640 | 0.647 | 0.692 | 0.656 | 0.690 | 0.656 |

**(b) Length = 64**

| Model | PPL | Ref | Lowercase | Zlib | Min-K% | Min-K%++ | DC-PDD | PAC | AECA |
|---|---|---|---|---|---|---|---|---|---|
| Mamba-1.4B | 0.582 | 0.588 | 0.566 | 0.604 | 0.615 | 0.663 | 0.651 | 0.652 | 0.650 |
| GPT-Neo-2.7B | 0.582 | 0.584 | 0.575 | 0.607 | 0.609 | 0.656 | 0.649 | 0.642 | 0.645 |
| Pythia-2.8B | 0.585 | 0.616 | 0.572 | 0.608 | 0.617 | 0.646 | 0.653 | 0.651 | 0.645 |
| OPT-6.7B | 0.573 | 0.645 | 0.587 | 0.598 | 0.595 | 0.641 | 0.634 | 0.623 | 0.634 |
| Pythia-6.9B | 0.616 | 0.650 | 0.589 | 0.636 | 0.662 | 0.722 | 0.683 | 0.698 | 0.694 |
| Pythia-12B | 0.625 | 0.651 | 0.612 | 0.643 | 0.668 | 0.732 | 0.684 | 0.693 | 0.699 |
| LLaMA-13B | 0.620 | 0.667 | 0.595 | 0.639 | 0.639 | 0.814 | 0.643 | 0.667 | 0.658 |
| GPT-NeoX-20B | 0.653 | 0.637 | 0.643 | 0.668 | 0.696 | 0.708 | 0.730 | 0.737 | 0.721 |
| Average | 0.605 | 0.630 | 0.592 | 0.625 | 0.638 | 0.698 | 0.666 | 0.670 | 0.668 |

**(c) Length = 128**

| Model | PPL | Ref | Lowercase | Zlib | Min-K% | Min-K%++ | DC-PDD | PAC | AECA |
|---|---|---|---|---|---|---|---|---|---|
| Mamba-1.4B | 0.636 | 0.599 | 0.584 | 0.658 | 0.664 | 0.672 | 0.666 | 0.673 | 0.682 |
| GPT-Neo-2.7B | 0.636 | 0.585 | 0.601 | 0.662 | 0.664 | 0.652 | 0.669 | 0.647 | 0.698 |
| Pythia-2.8B | 0.632 | 0.623 | 0.597 | 0.653 | 0.667 | 0.643 | 0.677 | 0.679 | 0.683 |
| OPT-6.7B | 0.622 | 0.632 | 0.580 | 0.640 | 0.663 | 0.674 | 0.670 | 0.618 | 0.686 |
| Pythia-6.9B | 0.659 | 0.664 | 0.608 | 0.681 | 0.700 | 0.696 | 0.688 | 0.709 | 0.718 |
| Pythia-12B | 0.660 | 0.657 | 0.619 | 0.682 | 0.706 | 0.722 | 0.688 | 0.710 | 0.716 |
| LLaMA-13B | 0.664 | 0.637 | 0.596 | 0.685 | 0.702 | 0.817 | 0.682 | 0.669 | 0.696 |
| GPT-NeoX-20B | 0.699 | 0.680 | 0.658 | 0.717 | 0.742 | 0.691 | 0.746 | 0.735 | 0.751 |
| Average | 0.651 | 0.635 | 0.605 | 0.672 | 0.689 | 0.696 | 0.686 | 0.680 | 0.704 |

**(d) Length = 256**

| Model | PPL | Ref | Lowercase | Zlib | Min-K% | Min-K%++ | DC-PDD | PAC | AECA |
|---|---|---|---|---|---|---|---|---|---|
| Mamba-1.4B | 0.666 | 0.584 | 0.588 | 0.679 | 0.701 | 0.637 | 0.626 | 0.689 | 0.707 |
| GPT-Neo-2.7B | 0.674 | 0.623 | 0.629 | 0.686 | 0.706 | 0.677 | 0.670 | 0.677 | 0.715 |
| Pythia-2.8B | 0.679 | 0.640 | 0.629 | 0.693 | 0.705 | 0.610 | 0.668 | 0.715 | 0.721 |
| OPT-6.7B | 0.639 | 0.691 | 0.612 | 0.662 | 0.669 | 0.626 | 0.625 | 0.658 | 0.696 |
| Pythia-6.9B | 0.692 | 0.649 | 0.592 | 0.703 | 0.710 | 0.608 | 0.664 | 0.711 | 0.713 |
| Pythia-12B | 0.688 | 0.646 | 0.633 | 0.706 | 0.721 | 0.643 | 0.687 | 0.716 | 0.729 |
| LLaMA-13B | 0.707 | 0.678 | 0.688 | 0.730 | 0.720 | 0.813 | 0.708 | 0.720 | 0.727 |
| GPT-NeoX-20B | 0.713 | 0.686 | 0.635 | 0.729 | 0.743 | 0.598 | 0.737 | 0.724 | 0.758 |
| Average | 0.682 | 0.650 | 0.626 | 0.699 | 0.709 | 0.652 | 0.673 | 0.701 | 0.721 |

*Table 11.* Full AUC results on the WikiMIA benchmark under the *paraphrased* setting.

**(a) Length = 32**

| Model | PPL | Ref | Lowercase | Zlib | Min-K% | Min-K%++ | DC-PDD | PAC | AECA |
|---|---|---|---|---|---|---|---|---|---|
| Mamba-1.4B | 0.621 | 0.588 | 0.606 | 0.620 | 0.640 | 0.667 | 0.611 | 0.686 | 0.657 |
| GPT-Neo-2.7B | 0.623 | 0.586 | 0.598 | 0.618 | 0.645 | 0.694 | 0.625 | 0.665 | 0.655 |
| Pythia-2.8B | 0.628 | 0.631 | 0.614 | 0.624 | 0.633 | 0.633 | 0.619 | 0.687 | 0.646 |
| OPT-6.7B | 0.616 | 0.668 | 0.598 | 0.613 | 0.628 | 0.671 | 0.633 | 0.664 | 0.643 |
| Pythia-6.9B | 0.657 | 0.658 | 0.631 | 0.648 | 0.689 | 0.727 | 0.664 | 0.715 | 0.692 |
| Pythia-12B | 0.666 | 0.662 | 0.655 | 0.657 | 0.686 | 0.709 | 0.681 | 0.726 | 0.691 |
| LLaMA-13B | 0.668 | 0.633 | 0.596 | 0.665 | 0.653 | 0.823 | 0.590 | 0.707 | 0.633 |
| GPT-NeoX-20B | 0.686 | 0.646 | 0.675 | 0.674 | 0.702 | 0.704 | 0.709 | 0.751 | 0.695 |
| Average | 0.646 | 0.634 | 0.622 | 0.640 | 0.660 | 0.704 | 0.642 | 0.700 | 0.664 |

**(b) Length = 64**

| Model | PPL | Ref | Lowercase | Zlib | Min-K% | Min-K%++ | DC-PDD | PAC | AECA |
|---|---|---|---|---|---|---|---|---|---|
| Mamba-1.4B | 0.577 | 0.573 | 0.574 | 0.583 | 0.615 | 0.659 | 0.623 | 0.655 | 0.659 |
| GPT-Neo-2.7B | 0.583 | 0.587 | 0.578 | 0.590 | 0.624 | 0.650 | 0.627 | 0.639 | 0.656 |
| Pythia-2.8B | 0.582 | 0.615 | 0.577 | 0.586 | 0.611 | 0.623 | 0.616 | 0.649 | 0.646 |
| OPT-6.7B | 0.570 | 0.641 | 0.584 | 0.578 | 0.596 | 0.646 | 0.607 | 0.616 | 0.637 |
| Pythia-6.9B | 0.613 | 0.653 | 0.595 | 0.615 | 0.655 | 0.714 | 0.657 | 0.694 | 0.692 |
| Pythia-12B | 0.622 | 0.651 | 0.614 | 0.622 | 0.668 | 0.725 | 0.681 | 0.692 | 0.703 |
| LLaMA-13B | 0.616 | 0.659 | 0.595 | 0.618 | 0.628 | 0.820 | 0.636 | 0.664 | 0.640 |
| GPT-NeoX-20B | 0.647 | 0.649 | 0.645 | 0.643 | 0.687 | 0.699 | 0.716 | 0.727 | 0.711 |
| Average | 0.601 | 0.628 | 0.595 | 0.604 | 0.636 | 0.692 | 0.645 | 0.667 | 0.668 |

**(c) Length = 128**

| Model | PPL | Ref | Lowercase | Zlib | Min-K% | Min-K%++ | DC-PDD | PAC | AECA |
|---|---|---|---|---|---|---|---|---|---|
| Mamba-1.4B | 0.631 | 0.609 | 0.591 | 0.612 | 0.676 | 0.676 | 0.666 | 0.683 | 0.692 |
| GPT-Neo-2.7B | 0.632 | 0.561 | 0.604 | 0.611 | 0.673 | 0.649 | 0.668 | 0.682 | 0.696 |
| Pythia-2.8B | 0.627 | 0.620 | 0.607 | 0.610 | 0.673 | 0.644 | 0.668 | 0.701 | 0.689 |
| OPT-6.7B | 0.613 | 0.637 | 0.586 | 0.598 | 0.665 | 0.680 | 0.671 | 0.641 | 0.688 |
| Pythia-6.9B | 0.657 | 0.662 | 0.619 | 0.635 | 0.701 | 0.697 | 0.701 | 0.714 | 0.726 |
| Pythia-12B | 0.659 | 0.658 | 0.624 | 0.638 | 0.713 | 0.724 | 0.704 | 0.730 | 0.734 |
| LLaMA-13B | 0.657 | 0.632 | 0.603 | 0.643 | 0.684 | 0.803 | 0.676 | 0.667 | 0.691 |
| GPT-NeoX-20B | 0.689 | 0.631 | 0.670 | 0.666 | 0.737 | 0.702 | 0.752 | 0.742 | 0.752 |
| Average | 0.646 | 0.626 | 0.613 | 0.627 | 0.690 | 0.697 | 0.688 | 0.695 | 0.709 |

**(d) Length = 256**

| Model | PPL | Ref | Lowercase | Zlib | Min-K% | Min-K%++ | DC-PDD | PAC | AECA |
|---|---|---|---|---|---|---|---|---|---|
| Mamba-1.4B | 0.637 | 0.614 | 0.622 | 0.526 | 0.684 | 0.603 | 0.618 | 0.705 | 0.710 |
| GPT-Neo-2.7B | 0.628 | 0.670 | 0.634 | 0.518 | 0.692 | 0.623 | 0.658 | 0.679 | 0.708 |
| Pythia-2.8B | 0.641 | 0.677 | 0.660 | 0.532 | 0.697 | 0.571 | 0.653 | 0.684 | 0.698 |
| OPT-6.7B | 0.596 | 0.634 | 0.611 | 0.505 | 0.633 | 0.595 | 0.598 | 0.646 | 0.662 |
| Pythia-6.9B | 0.658 | 0.694 | 0.633 | 0.553 | 0.700 | 0.617 | 0.682 | 0.728 | 0.727 |
| Pythia-12B | 0.653 | 0.691 | 0.653 | 0.548 | 0.683 | 0.600 | 0.677 | 0.691 | 0.698 |
| LLaMA-13B | 0.698 | 0.693 | 0.703 | 0.594 | 0.694 | 0.836 | 0.710 | 0.704 | 0.717 |
| GPT-NeoX-20B | 0.697 | 0.702 | 0.655 | 0.575 | 0.719 | 0.601 | 0.680 | 0.709 | 0.748 |
| Average | 0.651 | 0.672 | 0.646 | 0.544 | 0.688 | 0.631 | 0.659 | 0.693 | 0.709 |

*Table 12.* Average AUC results across three subsets (ArXiv, HackerNews, and PubMed Central) of the MIMIR benchmark. The values represent the mean performance of each method under specific Pythia model scales.

| Method | 160M | 1.4B | 2.8B | 6.9B | 12B | Avg. |
|---|---|---|---|---|---|---|
| PPL | 0.517 | 0.532 | 0.538 | 0.546 | 0.552 | 0.537 |
| Ref | 0.493 | 0.525 | 0.535 | 0.545 | 0.554 | 0.530 |
| Lowercase | 0.515 | 0.526 | 0.532 | 0.538 | 0.541 | 0.530 |
| Zlib | 0.521 | 0.532 | 0.537 | 0.542 | 0.547 | 0.536 |
| Min-K% | 0.518 | 0.531 | 0.540 | 0.550 | 0.560 | 0.540 |
| Min-K%++ | 0.515 | 0.527 | 0.538 | **0.551** | **0.564** | 0.539 |
| DC-PDD | 0.518 | 0.528 | 0.539 | 0.549 | 0.560 | 0.539 |
| PAC | 0.507 | 0.522 | 0.537 | 0.546 | 0.555 | 0.533 |
| AECA | **0.531** | **0.535** | **0.548** | 0.550 | **0.564** | **0.546** |

*Table 13.* Computational cost analysis: Average processing time per sample (in seconds) for different detection methods. The results highlight the computational efficiency of AECA compared to other baselines.

| | WikiMIA | | | | MIMIR | | | |
|---|---|---|---|---|---|---|---|---|
| Method | 32 | 64 | 128 | 256 | ArXiv | HackerNews | PubMed Central | Avg. |
| PPL | 0.114 | 0.135 | 0.192 | 0.355 | 0.189 | 0.209 | 0.204 | 0.200 |
| Ref | 0.151 | 0.183 | 0.294 | 0.482 | 0.182 | 0.186 | 0.186 | 0.238 |
| Lowercase | 0.226 | 0.274 | 0.445 | 0.717 | 0.337 | 0.345 | 0.343 | 0.384 |
| Zlib | 0.114 | 0.135 | 0.192 | 0.355 | 0.189 | 0.209 | 0.204 | 0.200 |
| Min-K% | 0.115 | 0.136 | 0.192 | 0.356 | 0.192 | 0.212 | 0.207 | 0.201 |
| Min-K%++ | 0.115 | 0.136 | 0.192 | 0.357 | 0.192 | 0.212 | 0.207 | 0.202 |
| DC-PDD | 0.100 | 0.144 | 0.234 | 0.386 | 0.158 | 0.160 | 0.220 | 0.200 |
| PAC | 0.671 | 0.860 | 1.308 | 2.200 | 0.801 | 0.954 | 0.841 | 1.091 |
| AECA | 0.100 | 0.155 | 0.210 | 0.331 | 0.166 | 0.171 | 0.222 | 0.194 |

*Table 14.* Specific ablation study results of the AECA method. We report the AUC scores for Pythia-2.8B and Pythia-6.9B on both the WikiMIA and MIMIR datasets. 'NLL Volatility' denotes analyzing solely the volatility of the raw loss sequence; '+ Calib. & Div.' represents the volatility divergence analysis incorporating Inherent Self-Information calibration; and '+ Conv.' further integrates convolution, corresponding to the complete AECA method.

| Configuration | WikiMIA(*Ori./Para.*) | | ArXiv | | HackerNews | | PubMed Central | |
|---|---|---|---|---|---|---|---|---|
| | 2.8B | 6.9B | 2.8B | 6.9B | 2.8B | 6.9B | 2.8B | 6.9B |
| NLL Volatility | 0.649/0.650 | 0.691/0.693 | 0.551 | 0.565 | 0.527 | 0.539 | 0.530 | 0.537 |
| + Calib. & Div. | 0.666/**0.673** | 0.690/0.697 | 0.564 | **0.573** | **0.535** | **0.544** | 0.534 | **0.540** |
| + Conv. (AECA) | **0.672**/0.670 | **0.704/0.709** | **0.569** | 0.571 | 0.531 | 0.541 | **0.544** | 0.539 |

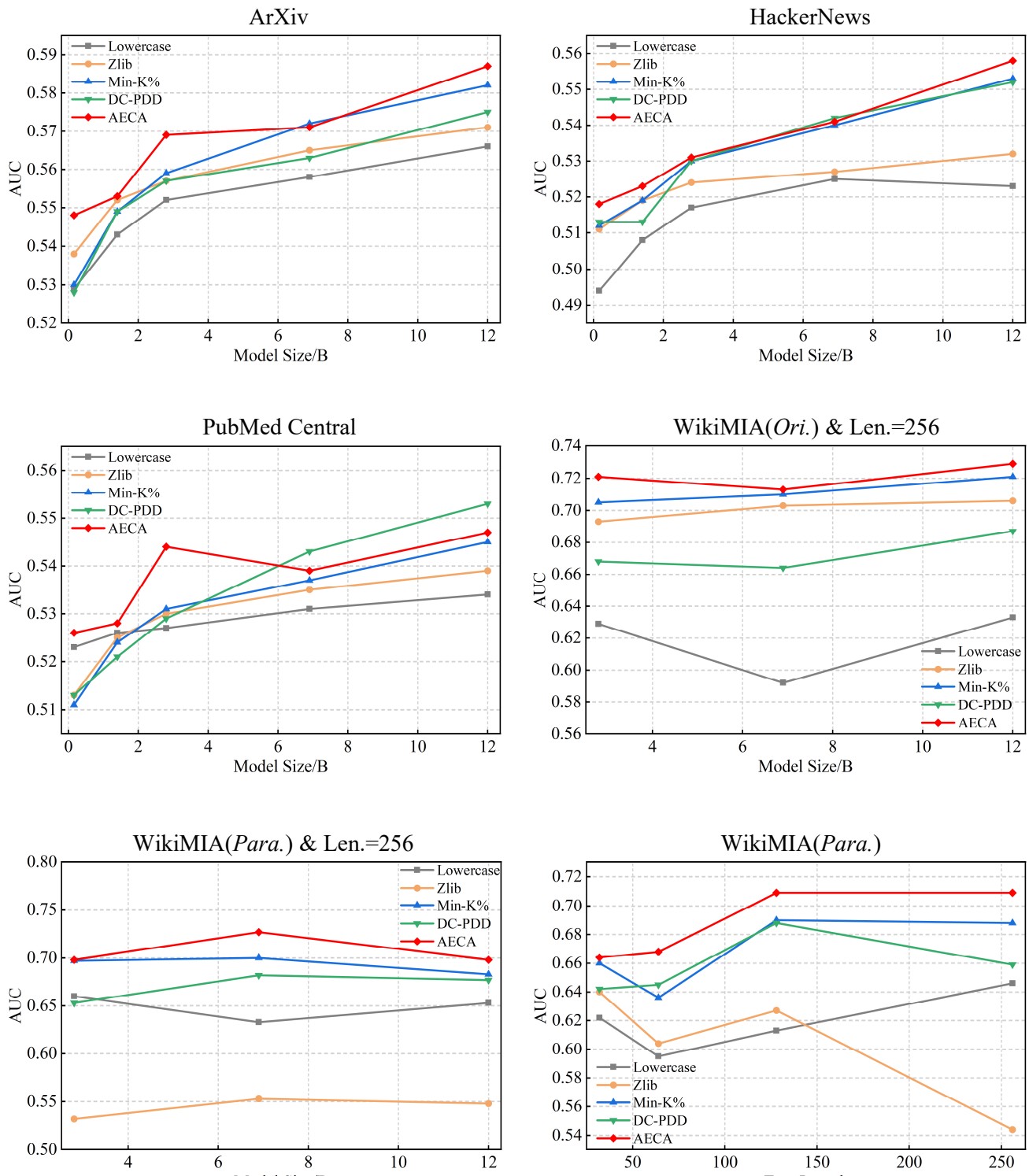

*Figure 5.* The performance of AECA across different model sizes and text lengths.

