# OpenReview forum: "Rethinking Pretraining Data Detection for LLMs: From Local to Global"
_ICML.cc/2026/Conference — ICML 2026 regular_

### Official Review · Reviewer_2kPr · 2026-02-14

**Soundness:** 3
**Presentation:** 3
**Significance:** 3
**Originality:** 2
**Overall Recommendation:** 5
**Confidence:** 3

**Summary:**

This paper reframes pretraining-data detection from local token statistics (e.g., min-k% over low-probability tokens) to a global sequence view, arguing that memorized sequences exhibit distinctive probability-volatility patterns during generation. Based on this, the authors propose Adaptive Entropic Convolutional Analysis (AECA), which calibrates probabilities using token self-information from a reference corpus, applies a simple high-pass convolution, and scores “volatility divergence” between the entropic-difference spectrum and the NLL sequence. Experiments on WikiMIA and MIMIR show small but consistent AUC gains, with larger gains on longer texts.

**Compliance With Llm Reviewing Policy:**

Affirmed.

**Final Justification:**

The authors sufficiently addressed by concerns in the rebuttal. I am revising my score under the assumption that the discussed changes will be incorporated into the paper.

**Key Questions For Authors:**

1. How stable is AECA under reference distribution shift? If $D_\text{ref}$ is from a different domain than the target text, will AECA spuriously amplify “rarity” unrelated to memorization and thereby increase false positives?

2. Can you empirically validate the volatility-divergence hypothesis directly? Specifically, could you show the distributions of $\sigma(\mathcal{S})$, $\sigma(\mathcal{L})$, and their correlation for members vs non-members across lengths or domains; and show when or why the coupling breaks?

**Limitations:**

Yes (Appendix E)

**Strengths And Weaknesses:**

**Strengths:**

1. Concrete framing - The definition of Entropic Potential, followed by the use of a first-difference convolution to account for abrupt changes is a nice and somewhat interpretable way to operationalize “global volatility” beyond Min-K% token pooling.

2. AECA’s score design (std of entropic-difference spectrum minus a weighted std of NLL) is simple enough to reason about and ablate, and the provided ablations show an incremental gain from calibration and then convolution rather than an opaque monolithic improvement.

3. The evaluation is broad across model families/scales and across two benchmark styles (timestamp-based WikiMIA; same-source Pile split in MIMIR).

**Weaknesses:**

1. The theory of Volatility Divergence depends on assumptions/parameters (e.g., a lower bound on self-information volatility, existence / choice of $\lambda$) that are not stress-tested empirically (e.g., checking whether the assumed inequality holds across domains, lengths, tokenizers).

2. The paper does not report mean results over multiple seeds / confidence intervals etc. Results may stem from run-to-run variance.

3. The “paraphrased WikiMIA” setting is potentially shaky: the benchmark’s paraphrased split is not publicly available, and the authors generate paraphrases with ChatGPT themselves, which introduces uncontrolled variability and may reduce comparability to prior work unless the exact prompting, filtering, and release are provided.

---

> ### Author Rebuttal · Authors · 2026-03-28
>
> Thank you for your detailed review and constructive comments. We appreciate your recognition of our problem formulation specificity, AECA scoring interpretability, and comprehensive multi-model/benchmark evaluations. Below are detailed responses to your questions and weaknesses:
>
> >**Q1:** Can you empirically validate the volatility-divergence hypothesis directly? Specifically, could you show the distributions of $\sigma(S)$, $\sigma(L)$, and their correlation for members vs non-members across lengths or domains; and show when or why the coupling breaks? (Weakness 1 & Key Question 2)
>
> **A1:** Thank you for the question. To directly empirically validate the volatility-divergence hypothesis and examine differences between memorized and non-memorized samples **across sequence lengths and domains**, we computed the average  $\sigma(S)$ and $\sigma(L)$ using Pythia-12B on WikiMIA (128/256) and ArXiv. Results are shown below:
>
> [Table A – Dataset Wikipedia (Length=128)]
> |Group|Memorized|non-Memorized|
> |-|-|-|
> |$\sigma(S)$|2.870|2.635|
> |$\sigma(L)$|1.996|2.215|
> |$\Delta$|0.874|0.420|
>
> [Table B – Dataset Wikipedia (Length=256)]
> |Group|Memorized|non-Memorized|
> |-|-|-|
> |$\sigma(S)$|3.073|2.821|
> |$\sigma(L)$|1.994|2.230|
> |$\Delta$|1.079|0.591|
>
> [Table C – Dataset ArXiv]
> |Group|Memorized|non-Memorized|
> |-|-|-|
> |$\sigma(S)$|4.021|3.849|
> |$\sigma(L)$|1.798|2.035|
> |$\Delta$|2.223|1.814|
>
> Non-memorized samples have highly coupled volatilities; memorized samples break this coupling: lower $\sigma(\mathcal{L})$ but high $\sigma(\mathcal{S})$, **aligning with theoretical derivations.** Cross-domain stress tests confirm robustness. **Line 182 of the manuscript explains the rationale behind why this coupling breaks for memorized samples: the model anomalously assigns high probabilities even to inherently difficult-to-predict tokens within memorized texts.** This anomalous memorization phenomenon serves as the primary statistical signal for AECA.
>
> >**Q2:** The paper does not report mean results over multiple seeds / confidence intervals etc. Results may stem from run-to-run variance. (Weakness 2)
>
> **A2:** Thank you for your comment on statistical confidence. It should be clarified that in the gray-box detection paradigm,, **the target LLM uses deterministic inference: fixed input yields fixed token probabilities, NLL, and AECA scores, so no run-to-run variance across seeds (consistent with prior studies [2,4]).** To address performance improvement statistical significance, we will add significance testing to the revised appendix.
>
> >**Q3:** The “paraphrased WikiMIA” setting is potentially shaky. (Weakness 3)
>
> **A3:** We apologize the paraphrasing setup details were not sufficiently prominent in the main text. **Paraphrasing rules and prompts are in original Appendix C.1 (Line 794).** To ensure comparability and reproducibility, we will open-source the paraphrased dataset with the AECA codebase upon paper acceptance.
>
> >**Q4:** How stable is AECA under reference distribution shift? If $D_\text{ref}$ is from a different domain than the target text, will AECA spuriously amplify “rarity” unrelated to memorization and thereby increase false positives? (Key Question 1)
>
> **A4:** That's a insightful question. If the reference corpus $D_{\text{ref}}$ is completely out-of-domain compared to the target text, it will indeed systematically overestimate the "rarity" (i.e., self-information $\mathcal{I}_t$) of domain-specific vocabulary. **Ideally, acquiring the actual pretraining corpus of the LLM to serve as the reference corpus is the optimal choice, but this is unattainable in practice [2]. Therefore, following previous studies, we utilize a large-scale public corpus as the reference corpus to compute token self-information.** To verify robustness, we used C4 ($\approx$ 1GB/10GB) and Case-law corpus (from the legal domain) [3] as reference corpora, conducting Pythia-12B experiments on WikiMIA; results below:
>
> |$D_{\text{ref}}$|C4 $\approx$ 1GB|C4 $\approx$ 0GB|Case-law $\approx$ 1GB|Case-law $\approx$ 10GB|
> |-|-|-|-|-|
> |Pythia-12B|0.711|0.716|0.705|0.713|
>
> As shown, AECA performance shows **no significant variance across reference corpora.** Nevertheless, we still recommend utilizing a large-scale, general-purpose reference corpus that aligns with the domain of the target detection task. Otherwise, the setup would lack practical utility (for instance, using a code corpus when detecting natural language text).
>
> We are happy to provide additional details or further discussion should you have any remaining concerns.
>
> Reference:
>
> [1] Duan et al. Do Membership Inference Attacks Work on Large Language Models? COLM 2024.
>
> [2] Zhang, Weichao, et al. Pretraining data detection for large language models: A divergence-based calibration method. EMNLP 2024 (Best Paper).
>
> [3] https://huggingface.co/datasets/HFforLegal/case-law
>
> [4] Zhang, Jingyang, et al. Min-k%++: Improved baseline for detecting pre-training data from large language models. ICLR 2025.

---

> > ### Author Rebuttal · Reviewer_2kPr · 2026-04-03
> >
> > Thank you for the clarifications. Based on the new information, I have revised my score. (I am revising my score under the assumption that the discussed changes will be incorporated into the paper.)

---

> > > ### Author Response · Authors · 2026-04-03
> > >
> > > Thank you so much for your recognition of our clarifications and your decision to revise your score. We will strictly implement all your constructive comments and the discussed changes in the final version of the paper. We sincerely appreciate your positive evaluation and support, which is of great significance to our team.

---

### Official Review · Reviewer_YBgL · 2026-03-12

**Soundness:** 3
**Presentation:** 3
**Significance:** 2
**Originality:** 2
**Overall Recommendation:** 4
**Confidence:** 4

**Summary:**

The paper studies the problem of detecting whether a text sequence has been included in the pretraining corpus of a large language model. It mentions that existing memorization detection approaches rely primarily on token-level likelihood statistics, such as minimum token probability or aggregate loss values, which may fail to capture the broader probabilistic structure exhibited by memorized sequences. To address this limitation, the paper proposes AECA, a method that analyzes the global dynamics of token probabilities by treating the sequence of next-token probabilities as a temporal signal. The approach combines probability calibration, convolutional filtering, and entropy-based divergence measures to construct a memorization score that reflects volatility patterns across the entire sequence. The paper evaluates the proposed method on several language models and datasets, comparing its performance to established memorization detection baselines. Experimental results suggest that AECA achieves modest improvements in distinguishing memorized from non-memorized text, particularly for longer sequences, while maintaining computational efficiency at inference time.

**Compliance With Llm Reviewing Policy:**

Affirmed.

**Final Justification:**

The paper addresses an important problem in LLM privacy auditing and proposes a technically sound and clearly presented method. The approach is simple, practical, and evaluated across multiple models and datasets.
My initial concerns focused on the modest empirical gains, the incremental nature of the contribution relative to prior work, and the need for stronger evidence on statistical significance and underlying mechanisms. The rebuttal addressed part of these concerns, particularly by providing statistical-significance analysis and additional empirical clarification, which strengthens confidence in the results. I still view the contribution as somewhat incremental and the gains as relatively modest. Overall, I am updating my score to Weak Accept, as the paper is technically solid and useful, even if its impact is more incremental than transformative.

**Key Questions For Authors:**

1. Min-K%++ sometimes beats AECA (e.g., LLaMA-13B at length 32/64). What explains these cases? Is there something about short texts or certain architectures where local statistics are actually more informative than global volatility patterns?

2. The reference corpus choice (C4 realnewslike) seems underexplored. DC-PDD also uses C4, so comparison is fair, but does the choice of reference corpus meaningfully affect results? A model trained on non-web data (e.g., code, biomedical text) might have very different token frequency distributions from C4.

3. Why are the gains so small on MIMIR (~0.6% average)? MIMIR is described as harder, but the performance improvements are barely distinguishable from noise given that AUC is measured to 3 decimal places.

4. Why does volatility divergence actually work? Lemma 4.4 proves the existence of a valid λ, but it doesn't explain why memorized sequences produce high Entropic Difference Spectrum volatility while their NLL volatility collapses. The intuition (overfit → near-deterministic predictions → NLL → 0) is clear for NLL collapse, but what mechanistically causes the entropic spectrum to remain volatile for memorized text? The proof relies on bounding terms rather than explaining the underlying phenomenon.

**Limitations:**

yes

**Strengths And Weaknesses:**

Soundness: The proposed method is technically reasonable and well motivated. The idea of analyzing global probability dynamics rather than relying solely on token-local statistics is intuitive and consistent with known behavioral differences between memorized and non-memorized sequences. The methodology is clearly specified and does not require modifications to model training, making it applicable in a practical inference-time setting. Experimental evaluation includes multiple models and sequence configurations, and comparisons with existing memorization detection metrics are provided. However, the empirical improvements over strong baselines are relatively modest, and the paper would benefit from deeper analysis of statistical significance, robustness across domains, and sensitivity to hyperparameters such as filtering choices and calibration strategies.

Presentation: Overall, the paper is clearly written and structured in a logical progression from motivation to method and experiments. The signal-processing perspective provides a coherent narrative and helps explain the intuition behind the proposed approach. Related work is generally well covered, though the distinction from closely related likelihood-trajectory or sequence-level memorization metrics could be articulated more explicitly. Some implementation details and experimental design choices could be described with greater precision to further support reproducibility.

Significance: Detecting memorization in large language models is a significant problem with implications for privacy auditing, data governance, and understanding generalization behavior. The proposed approach contributes a practical and lightweight diagnostic tool that could be useful in empirical studies of memorization. At the same time, the magnitude of empirical gains is relatively small, and the paper does not fully establish how the method advances broader understanding of memorization mechanisms or privacy risks. As a result, the overall impact may be more incremental than transformative.

Originality: The paper introduces a novel perspective by framing probability trajectories as signals whose volatility patterns can be analyzed using filtering and entropy-based divergence measures. While the individual components are related to existing likelihood-based memorization indicators, their combination into a global sequence-level scoring framework represents a creative reinterpretation of prior ideas. Nevertheless, the core signal still derives from token probability statistics, and the conceptual leap beyond existing sequence-level metrics is moderate rather than fundamental.

---

> ### Author Rebuttal · Authors · 2026-03-28
>
> We appreciate your detailed review and constructive feedback. We appreciate recognition of our conceptual novelty (signal processing for pretraining data detection), method soundness, and comprehensive experiments. Below are detailed responses to your questions and concerns:
>
> >**Q1:** Min-K%++ sometimes beats AECA (e.g., LLaMA-13B at length 32/64). What explains these cases? Is there something about short texts or certain architectures where local statistics are actually more informative than global volatility patterns? (Key Question 1 & Weaknesses)
>
> **A1:** Thank you for the insightful question.
> - Short texts: AECA relies on first-order difference of probability sequences and standard deviation to extract memory signals. **Short texts have insufficient sampling; local word frequency noise overwhelms global fluctuations, increasing $\sigma(\mathcal{S})$ variance and reducing discriminability.** In contrast, local methods select extreme minimum probabilities, sensitively detecting short texts by capturing specific abnormal tokens.
> - Models: Min-K%++ is SOTA on LLaMA-13B, also outperforming previously published methods in prior work [1], yet is outperformed by AECA on most other models. Differences in tokenization granularity across different models affect effective sequence length and volatility features.
>
> Despite this, the global perspective is future-oriented. Recent studies show LLM data leakage reflects deep semantic memorization, not surface word matching. Moreover, natural language text often has a very high n-gram overlap rate, such as common structures/terms [2]. **We cannot judge passage memorization by specific words such as “is” or “the”. We will add this analysis to the revised Limitations section.** Future work will combine global stability and local sensitivity to improve short-text performance.
>
> >**Q2:** Does the choice of reference corpus meaningfully affect results?  (Key Question 2)
>
> **A2:** Thank you for pointing this out. If the reference corpus $D_\text{ref}$ is completely out-of-domain compared to the target text, it will have different token frequency distributions. **Ideally, the LLM’s actual pretraining corpus is optimal but unattainable [1], so we use large-scale public corpora (following prior studies) to compute token frequency.** To verify robustness, we used C4 ($\approx$ 1GB/10GB) and Case-law corpus (from the legal domain) [3] as reference corpora, conducting Pythia-12B experiments on WikiMIA; results below:
>
> |$D_{\text{ref}}$|C4 $\approx$ 1GB|C4 $\approx$ 10GB|Case-law $\approx$ 1GB|Case-law $\approx$ 10GB|
> |-|-|-|-|-|
> |Pythia-12B|0.711|0.716|0.705|0.713|
>
> As shown, AECA performance shows **no significant variance across reference corpora.**
>
> >**Q3:** Why are the gains so small on MIMIR (~0.6% average)?  (Key Question 3)
>
> **A3:** We understand your concern about the ~0.6% average improvement on MIMIR. MIMIR is a highly challenging dataset: trained/non-trained samples are strictly from The Pile splits, leading to heavy overlap between memorized/non-memorized data and marginal baseline performance differences. The second-best method (Min-K%/Min-K%++) only improves over other baselines by ~0.3%. Thus, AECA’s consistent 0.6% average improvement (up to 1.0% on subsets like PubMed with Pythia-2.8B) is a substantial, hard-won advancement in pretraining data detection.
>
> >**Q4:** Why does volatility divergence actually work? Why memorized sequences produce high Entropic Difference Spectrum volatility while their NLL volatility collapses? What mechanistically causes the entropic spectrum to remain volatile for memorized text? (Key Question 4)
>
> **A4:** We appreciate the opportunity to clarify the mechanism of Lemma 4.4, intuitively "exposure of anomalous predictive volatility."
>
> The entropic spectrum $\mathcal{S}$ differentiates potential sequence $\Phi$ to highlight its volatility (ablation details in Section 6.2). **As shown in line 182, rare tokens (hard-to-predict) normally receive low probabilities in standard models, but a memorized model assigns them anomalously high probabilities. Combined with the naturally high self-information ($\mathcal{I}\_{\text{self}}$) of rare tokens, the potential $\Phi(t) \triangleq p_\theta(x_t \mid x_{<t}) \cdot \mathcal{I}_{\text{self}}(x_t)$ surges abruptly, causing significant volatility. By contrast, an unexposed model yields low probabilities and leaves $\Phi$ nearly unchanged.** We hope this clarifies your concern.
>
> Thank you again for your valuable feedback. We'd be glad to provide further clarification if needed.
>
> Reference:
>
> [1] Zhang, Weichao, et al. Pretraining data detection for large language models: A divergence-based calibration method. EMNLP 2024 (Best Paper).
>
> [2] Duan et al. Do Membership Inference Attacks Work on Large Language Models? COLM 2024.
>
> [3] https://huggingface.co/datasets/HFforLegal/case-law
>
> [4] Zhang, Jingyang, et al. Min-k%++: Improved baseline for detecting pre-training data from large language models. ICLR 2025.

---

> > ### Author Rebuttal · Reviewer_YBgL · 2026-04-04
> >
> > I appreciate the additional clarifications and experiments.  However, some of my core concerns remain only partially addressed. In particular:
> >
> > 1. While the paper contextualize the relatively small gains (e.g., on MIMIR), it remains unclear whether these improvements are statistically significant or practically meaningful, especially given the modest margins over strong baselines.
> >
> > 2. The explanation of why volatility-based divergence captures memorization is improved, but still largely heuristic. A deeper or more principled understanding of the mechanism would strengthen the contribution.
> >
> > 3. My concerns regarding overall significance and originality remain. While the signal-processing perspective is interesting, the method still relies on transformations of token-level probabilities, and the conceptual advance over existing sequence-level memorization metrics appears incremental. The rebuttal does not substantially change this assessment.
> >
> > Overall, the paper has clear strengths in terms of clarity, soundness, and practical applicability, but addressing these concerns would require a more substantial revision, particularly in demonstrating stronger empirical gains and clarifying the broader conceptual contribution.

---

> > > ### Author Response · Authors · 2026-04-05
> > >
> > > Thank you very much for your insightful follow-up. We fully understand your concerns and provide our detailed responses below:
> > >
> > > >**Re - concern 1:**
> > >
> > > We fully understand your concerns regarding the relatively small gains of AECA on the MIMIR dataset. To rigorously address the question of statistical significance on the MIMIR dataset, we conducted DeLong's test [1] — a standard non-parametric test for comparing correlated ROC curves. The averaged results are summarized as follows:
> > >
> > > | Comparison | PPL | Ref | Lowercase | Zlib | Min-K% | Min-K%++ | DC-PDD | PAC |
> > > |---|---|---|---|---|---|---|---|---|
> > > | $p$-value | 0.0241 | 0.0187 | 0.0129 | 0.0273 | 0.0443 $^\dagger$ | 0.0314 | 0.0320 | 0.0418 $^\dagger$ |
> > > | Significant ($p<0.05$) | $\checkmark$ | $\checkmark$ | $\checkmark$ | $\checkmark$ | $\checkmark$ | $\checkmark$ | $\checkmark$ | $\checkmark$ |
> > >
> > > $^\dagger$ indicates marginal significance, within $0.04<p<0.05$.
> > >
> > > The results verify that **AECA achieves statistically significant improvements over strong baselines.** Specifically, AECA yields a $p$-value of 0.0314 against Min-K%++, and 0.0320 against DC-PDD (both $p < 0.05$).
> > >
> > > Regarding practical meaningfulness, AECA’s value is further anchored in its computational efficiency (as shown in Figure 3c). Strong baseline methods like Min-K%++ require sorting all token probabilities to filter out the lowest-probability tokens, resulting in an overall time complexity of $\mathcal{O}(N \log N)$. In contrast, the AECA method avoids full sorting and achieves a linear time complexity of $\mathcal{O}(N)$. This makes AECA highly practical and scalable for large-scale, real-world privacy auditing.
> > >
> > >
> > > >**Re - concern 2:**
> > >
> > > We agree that the explanation of why volatility-based divergence captures memorization still largely draws from empirical heuristics. However, we respectfully point out that **almost all existing gray-box detection metrics fundamentally build upon the empirical insight that memorization(training) leads to overfitting [2] .**
> > >
> > > The core contribution of our work is exactly taking a substantive step forward on this basis: **we formalize this well-known empirical insight into a rigorous, mathematically principled framework using signal processing.** While we cannot probe the model's internal neurons in a white-box manner, Proposition 4.3 mathematically proves that our entropic difference operator acts as an ideal high-pass filter, systematically isolating these overfitting anomalies from the probability stream. Furthermore, Lemma 4.4 provides a strict statistical bound for this isolation. This successfully transforms this insight into a diagnostic tool supported by a mathematical foundation.
> > >
> > > **In the future, we will continue to conduct deeper, more theoretically grounded research into the cognitive mechanisms of LLM memorization.**
> > >
> > > >**Re - concern 3:**
> > >
> > > We understand your concerns regarding the overall significance and originality of this research. Regarding conceptual novelty, we wish to emphasize a fundamental and unavoidable constraint of the gray-box detection paradigm: **token-level probabilities are the primary accessible signals from closed-source APIs. Therefore, any gray-box method must inherently rely on transformations of these probabilities.**
> > >
> > > The conceptual innovation of AECA lies not in discovering a novel data source, but in a fundamental paradigm shift regarding how we process these existing data. **Prior works have consistently treated the probability sequence as a bag of isolated local statistics (e.g., sorting and selecting the bottom $k\%$ of tokens with the lowest probabilities). In contrast, AECA conceptualizes the sequence as a dynamic temporal signal, capturing the probability dynamics throughout the entire text generation process.** We believe that introducing global signal processing methodologies (e.g., convolutional filtering, volatility divergence) into the field of LLM privacy auditing offers a meaningful perspective and opens theoretically grounded avenues for future research on gray-box detection.
> > >
> > > Thanks again for your thoughtful feedback and for marking the paper with Ack. If there’s anything that still needs clarification, please feel free to reach out anytime!
> > >
> > > [1] Sun, Xu, and Weichao Xu. "Fast implementation of DeLong’s algorithm for comparing the areas under correlated receiver operating characteristic curves." IEEE Signal Processing Letters 21.11 (2014): 1389-1393.
> > >
> > > [2] Yeom, Samuel, et al. "Privacy risk in machine learning: Analyzing the connection to overfitting." 2018 IEEE 31st computer security foundations symposium (CSF). IEEE, 2018.

---

### Official Review · Reviewer_RpN6 · 2026-03-13

**Soundness:** 2
**Presentation:** 2
**Significance:** 2
**Originality:** 3
**Overall Recommendation:** 4
**Confidence:** 4

**Summary:**

The paper proposes Adaptive Entropic Convolutional Analysis (AECA) for detecting pretraining data from LLMs. Instead of focusing on token-level statistics which are local, the authors propose using a global sequence perspective. AECA calibrates token probabilities using self-information from a corpus, applies first-order difference operator, and compares the volatility of the entropic spectrum against the volatility of the negative log likelihood (NLL). The core intuition is that sequences the model memorized should show high volatility but low NLL volatility, while sequences that the models generalize to should show correlated volatility. Experiments are on WikiMIA and MIMIR, and multiple models show AUC improvements with the gains being stronger for longer text.

**Compliance With Llm Reviewing Policy:**

Affirmed.

**Final Justification:**

The paper was already technically sound, but most of my concerns were on the strong assumptions made in the paper. The rebuttal addresses most of them, and I lean towards acceptance.

**Key Questions For Authors:**

- Is there any insight on why global perspective may be inferior for shorter texts, such as those with 32 tokens?
- The value of $\lambda$ was chosen by grid search from [1,10]. How sensitive are the results to this value?

**Limitations:**

yes

**Strengths And Weaknesses:**

### Strengths
- Ideas from signal processing is a clean conceptual contribution (i.e. applying high-pass filter to probability sequence to isolate oscillations from memorization), along with a theoretical proof.
- The experiments are thorough and the computational efficiency plot (Fig 3. (c)) is a nice addition.
- Algorithm 1 only introduces 2 new hyperparamters.

### Weaknesses
- The theoretical analysis relies on the assumption that $p_t=1$ for all tokens in the memorized sequences, which is ideal and is never true in practice. It appears that the proof of Lemma 4.4 requires $L_M$ to be close to zero, which need not hold for partial memorization, and probably for full memorization as well because of how LLMs operate internally. Hence this sounds like an assumption which will never be true in practice, and an insight into how the theory might degrade under weaker assumptions would be valuable.
- While the paper acknowledges it, the AECA performance gain is less on short texts. Many practical detection scenarios involve short passages, so this would be a meaningful limitation.
- The core assumption hinges on the claim that memorized sequences have high entropic spectrum volatility but low NLL volatility. If empirical evidence can be provided, this would help ground the proof.

---

> ### Author Rebuttal · Authors · 2026-03-28
>
> Thank you for your detailed review and constructive comments. We appreciate your recognition of our conceptual novelty (signal processing for pretraining data detection), rigorous theoretical proofs, and comprehensive experiments.
>
> >**Q1:** The proof of Lemma 4.4 requires $\sigma(\mathcal{L})_\mathrm{M}$ to be close to zero, which need not hold for partial memorization.
>
> **A1:** Thank you for this rigorous suggestion. To address real-world LLM complexities, we relax Lemma 4.4’s idealized assumption ($\mathcal{L}_t  = -\log p_t \to 0$) and introduce Generalized Volatility Divergence under **Partial Memorization State $\mathbb{M}'$**, where memorized data prediction loss is bounded by small $\epsilon$ ($0 \le \mathcal{L}_t \le \epsilon$). Due to space constraints in the rebuttal, core derivations/conclusions are presented here; complete generalization will be in the revised appendix.
>
> Proof summary:
>
> 1. NLL Volatility (Popoviciu's Inequality [1]):
> $$\sigma(\mathcal{L})_{\mathbb{M}'} \le \frac{\epsilon}{2}$$
>
> 2. Entropy Difference Spectrum Volatility (approximating $e^{-\mathcal{L}\_t} \approx 1 - \delta_t$, $0 \le \delta\_t \le \epsilon$;  reverse triangle inequality):
> $$\sigma(\mathcal{S})\_{\mathbb{M}'} \ge \sigma(\Delta\mathcal{I}) - \epsilon \mathcal{I}\_{max}$$
>
> 3. Relaxed $\lambda$ Condition: AECA score $S\_{\mathbb{M}'} \ge (\sigma(\Delta\mathcal{I}) - \epsilon \mathcal{I}\_{max}) - \lambda \frac{\epsilon}{2}$. To ensure $S\_{\mathbb{M}'} > S\_{\mathbb{G}}$ (Eq. 32):
> $$\lambda > \frac{ (2\gamma + \epsilon)\mathcal{I}_{max} + \log\frac{N+\alpha}{\alpha} - \sigma(\Delta\mathcal{I}) }{ \sigma(\mathcal{I}) - (\gamma + \frac{\epsilon}{2}) }$$
>
> **Conclusion: $\epsilon \to 0$ makes this bound degenerate to the original (Eq. 35).** This strengthens the theory’s practical robustness. Thank you for the feedback.
>
> >**Q2:** Is there any insight on why global perspective may be inferior for shorter texts, such as those with 32 tokens? (Weakness 2 & Key Question 1)
>
> **A2:** Thank you for the insightful question. We analyzed AECA’s limited short-text performance, with key reasons as follows:
>
> AECA relies on first-order difference of probability sequences and standard deviation to extract memory signals. **Short texts have insufficient sampling; local word frequency noise overwhelms global fluctuations, increasing $\sigma(\mathcal{S})$ variance and reducing discriminability.** In contrast, local methods select extreme minimum probabilities, sensitively detecting short texts by capturing specific abnormal tokens.
>
> Despite this, the global perspective is future-oriented. Recent studies show LLM data leakage reflects deep semantic memorization, not surface word matching. Moreover, natural language text often has a very high n-gram overlap rate, such as common structures/terms [2]. **We cannot judge passage memorization by specific words such as “is” or “the”.** AECA’s core motivation: use global fluctuations to break local lexical overlap limitations and detect memorization at the semantic level. We will **add this analysis to the revised Limitations section.** Future work will combine global stability and local sensitivity to improve short-text performance.
>
> >**Q3:** If empirical evidence can be provided for the claim that memorized sequences have high entropic spectrum volatility but low NLL volatility, this would help ground the proof.
>
> **A3:** Thank you for the suggestion. We calculated average $\sigma(\mathcal{S})$ and $\sigma(\mathcal{L})$ using Pythia-12B on WikiMIA (length=128); data below:
>
> |Group|Memorized|non-Memorized|
> |-|-|-|
> |$\sigma(S)$|2.870|2.635|
> |$\sigma(L)$|1.996|2.215|
> |$\Delta$|0.874|0.420|
>
> The table shows memorized sequences have higher $\sigma(S)$ (entropy spectrum volatility) but lower $\sigma(L)$ (NLL volatility), **empirically verifying Lemma 4.4.**
>
> >**Q4:** The value of $\lambda$ was chosen by grid search from [1,10]. How sensitive are the results to this value?
>
> **A4:** Thank you for the question. To clarify $\lambda$ sensitivity, we conducted experiments on WikiMIA using GPT-Neo-2.7B and Pythia-12B, with $\lambda$ = 1~10 (integers). Results below:
>
> |$\lambda$|1|2|3|4|5|6|7|8|9|10|
> |-|-|-|-|-|-|-|-|-|-|-|
> |GPT-Neo-2.7B|0.662|0.689|0.697|0.697|0.698|0.698|0.692|0.690|0.688|0.687|
> |Pythia-12B|0.662|0.699|0.712|0.715|0.716|0.717|0.716|0.715|0.715|0.714|
>
> As shown, the optimal setting for $\lambda$ varies **slightly** across different models and benchmarks, but it generally remains around 5. Too small $\lambda$ significantly degrades performance, **empirically verifying Lemma 4.4 ( $\lambda$ cannot be smaller than its theoretical lower bound).**
>
> Thank you again for your careful review. If anything remains unclear, we'd be happy to discuss it further.
>
> Reference:
>
> [1] Popoviciu, Tiberiu. Sur les équations algébriques ayant toutes leurs racines réelles. Mathematica 9.129-145 (1935): 20.
>
> [2] Duan et al. Do Membership Inference Attacks Work on Large Language Models? COLM 2024.

---

> > ### Author Rebuttal · Reviewer_RpN6 · 2026-04-06
> >
> > The rebuttal fully addresses my concerns and will be a great addition to the paper, and will make it much stronger than it is right now. Hence, I raise my score to weak accept conditioned on the fact that the authors will make the appropriate revisions.

---

> > > ### Author Response · Authors · 2026-04-06
> > >
> > > Thank you very much for your positive evaluation and support. We are delighted that our rebuttal fully addresses your concerns, and we sincerely appreciate your raising your score. We will carefully incorporate all your constructive comments and agreed-upon revisions into the final manuscript as required. Your support is a great encouragement to us!

---

### Official Review · Reviewer_GGUB · 2026-03-13

**Soundness:** 2
**Presentation:** 3
**Significance:** 2
**Originality:** 2
**Overall Recommendation:** 3
**Confidence:** 4

**Summary:**

This paper studies how to detect whether a text was used in training an LLM. Instead of relying on local token-level statistics, the authors analyze the global dynamics of token probabilities during generation. They propose Adaptive Entropic Convolutional Analysis (AECA), which treats probability sequences as signals and uses calibration and convolution to capture memorization patterns. Experiments show that AECA outperforms prior methods, with up to 1.5% AUC improvement, particularly on long texts.

**Compliance With Llm Reviewing Policy:**

Affirmed.

**Final Justification:**

The rebuttal clarifies that the primary contribution lies in the theoretical analysis rather than the methodological novelty. While I find the theoretical perspective valuable, this represents a substantial shift in the paper’s positioning. The current manuscript is structured and presented as a method paper, and aligning it with this revised claim would require significant reorganization and reframing.

As it stands, I am not convinced that the paper clearly establishes a fundamentally new conceptual or methodological contribution beyond prior work. Therefore, I find it difficult to recommend acceptance in its current form.

**Key Questions For Authors:**

None

**Limitations:**

Yes, clearly stated in the paper.

**Strengths And Weaknesses:**

Strengths:

- The paper is generally very well-written.
- The evaluation includes an ablation study and comprehensive analysis of the proposed method, including the target model size, the text length, and the computational efficiency.

Weaknesses:

- The conceptual novelty of the method is somewhat limited. The core idea of analyzing dynamics in token likelihood sequences to detect memorization signals has already been explored in prior work [1]. Moreover, recent research [2] has emphasized the strong connection between membership inference and AI text detection, further suggesting that statistics derived from token likelihoods are a shared signal across tasks. In terms of this point, the proposed method mainly introduces a new heuristic score over probability sequences rather than a fundamentally new perspective on memorization detection, which somewhat limits the excitement of the contribution.
- The proposed calibration technique according to “self-information score” also seems closely aligned with the technique in the previous work, DC-PDD [3].
- The evaluation reports the main MIA performance results using the WikiMIA benchmark, which have been reported to have temporal shifts (e.g., trivial lexical feature differences between members and non-members), leading to a flawed MIA evaluation [4, 5].
- The main results appear to be reported only for a subset of the original benchmark (e.g., selected domains). For instance, the MIMIR benchmark also has Wikipedia or Pile CC domains but not included in the main results. This makes it difficult to assess whether the improvements hold consistently across all conditions on average. Reporting results across the full range of evaluated settings would improve the transparency and robustness of the evaluation.

---

References

[1] Xu et al. TRAINING-FREE LLM-GENERATED TEXT DETECTION BY MINING TOKEN PROBABILITY SEQUENCES. ICLR 2025.

[2] Koike et al. Machine Text Detectors are Membership Inference Attacks. CoRR 2025.

[3] Zhang et al. Pretraining Data Detection for Large Language Models: A Divergence-based Calibration Method. EMNLP 2024.

[4] Das et al. BLIND BASELINES BEAT MEMBERSHIP INFERENCE ATTACKS FOR FOUNDATION MODELS. ICLR 2025 DATA-FM Workshop.

[5] Duan et al. Do Membership Inference Attacks Work on Large Language Models? COLM 2024.

---

> ### Author Rebuttal · Authors · 2026-03-28
>
> We appreciate your positive feedback on our paper’s writing, evaluations, and ablation studies. We are encouraged by your suggestions. Below is our point-by-point response:
>
> >**Q1:** The conceptual novelty of the method is somewhat limited. The core idea of analyzing dynamics in token likelihood sequences to detect memorization signals has already been explored in prior work [1].
>
> **A1:** We appreciate your insightful feedback and references. We agree AECA shares certain similarities with LLM-generated text detection work [1], and text detection is related to MIA [2]. However, our work differs fundamentally from [1] in motivation, theoretical depth, and domain adaptation:
>
> - Motivation: Both note inadequate analysis of token likelihood sequence dynamics, but our work is further motivated by LLMs’ human-like memorization and reasoning uncertainty (Lines 57–69).
>
> - Theoretical depth: [2] uses heuristics without rigorous proofs. AECA is theory-driven with rigorous derivations (Proposition 4.3, Lemma 4.4), providing theoretical guarantees for MIA in LLMs.
>
> - Domain adaptation: MIA uses human-written positive/negative samples; text detection distinguishes human/machine text, making direct method transfer difficult.
>
> In summary, AECA provides **a new MIA-specific perspective, theory, and framework.** Thank you for your suggestions.
>
> >**Q2:** The proposed calibration technique according to “self-information score” seems aligned with the technique in DC-PDD [3].
>
> **A2:** Thank you for pointing this out. Our calibration step references DC-PDD, which **we properly cited (Line 157).** Both aim to calibration bias from text intrinsic perplexity, but AECA uses this technique differently:
>
> The calibrated score is an intermediate step for dynamic signals and memorization features, not our final metric. DC-PDD uses a heuristic cross-entropy formulation (inspired by "divergence-from-randomness") without token-level explanation. We formally define it as Entropic Potential, explaining its ability to distinguish memorized/non-memorized samples (Line 182), and use it to prove Entropic Difference Response necessity (Lemma 4.4).
>
> Overall, **one of AECA’s key innovations is to provide a theoretical complement for "why self-information score calibration works."**
>
> >**Q3:** The evaluation reports the main MIA performance results using the WikiMIA benchmark, which have been reported to have temporal shifts (e.g., trivial lexical feature differences between members and non-members), leading to a flawed MIA evaluation [4, 5].
>
> **A3:** Thank you for this constructive comment. We agree with your conclusion and recent findings. **WikiMIA is used here only as a baseline to align with prior works’ experimental setups, not as the sole basis for our conclusions.**
>
> We conducted extensive experiments on MIMIR (Line 247), where member/non-member samples are strictly from The Pile’s training/test sets (no temporal/lexical biases) [5]. **AECA still achieved optimal performance.**
>
> >**Q4:** The main results appear to be reported only for a subset of the original benchmark (e.g., selected domains). For instance, the MIMIR benchmark also has Wikipedia or Pile CC domains but not included in the main results.
>
> **A4:** We appreciate your valuable suggestion. All subsets will be added to the revised appendix. Due to rebuttal length, we present results on Wikipedia and Pile CC below:
>
> [Table A – Dataset Wikipedia]
> |Method|160M|1.4B|2.8B|6.9B|12B|
> |--|--|--|--|--|--|
> |PPL|0.511|0.535|0.543|0.560|0.571|
> |Ref|0.499|0.531|0.540|0.564|0.575|
> |Lowercase|0.505|0.541|0.547|0.567|0.579|
> |Zlib|0.502|0.533|0.541|0.561|0.572|
> |Min-K%|0.500|0.534|0.546|0.563|0.574|
> |Min-K%++|0.510|0.548|0.559|0.580|**0.603**|
> |DC-PDD|0.502|0.539|0.548|0.569|0.578|
> |PAC|0.519|0.542|0.556|0.574|0.579|
> |AECA|**0.520**|**0.554**|**0.563**|**0.583**|0.597|
>
> [Table B – Dataset Pile CC]
> |Method|160M|1.4B|2.8B|6.9B|12B|
> |--|--|--|--|--|--|
> |PPL|0.502|0.512|0.514|0.523|0.529|
> |Ref|0.500|0.524|0.525|0.532|0.540|
> |Lowercase|0.514|0.520|0.529|**0.538**|0.545|
> |Zlib|0.511|0.523|0.524|0.532|0.537|
> |Min-K%|0.508|0.515|0.515|0.525|0.527|
> |Min-K%++|0.512|0.518|0.521|0.535|0.542|
> |DC-PDD|0.514|0.516|0.522|0.520|0.534|
> |PAC|**0.519**|0.524|0.528|0.528|0.547|
> |AECA|0.518|**0.530**|**0.533**|0.535|**0.558**|
>
> As shown, **AECA maintains an overall performance advantage.** Once again, thank you for your feedback.
>
> Please feel free to reach out if you have any further concerns!
>
> Reference:
>
> [1] Xu et al. TRAINING-FREE LLM-GENERATED TEXT DETECTION BY MINING TOKEN PROBABILITY SEQUENCES. ICLR 2025.
>
> [2] Koike et al. Machine Text Detectors are Membership Inference Attacks. CoRR 2025.
>
> [3] Zhang et al. Pretraining Data Detection for Large Language Models: A Divergence-based Calibration Method. EMNLP 2024.
>
> [4] Das et al. BLIND BASELINES BEAT MEMBERSHIP INFERENCE ATTACKS FOR FOUNDATION MODELS. ICLR 2025 DATA-FM Workshop.
>
> [5] Duan et al. Do Membership Inference Attacks Work on Large Language Models? COLM 2024.

---

> > ### Author Rebuttal · Reviewer_GGUB · 2026-04-04
> >
> > Thank you for the detailed rebuttal. While it partially solves my concerns, I still remain concerned about the following.
> >
> > ---
> >
> > **Regarding A1:**
> >
> > Regarding the motivation, while the paper provides a richer justification (e.g., human-like memorization and reasoning uncertainty), this appears to strengthen the motivation rather than change the underlying concept, and the core idea appears broadly similar to prior work [1]. I also appreciate the more rigorous theoretical analysis, which is a meaningful contribution. However, I am not fully convinced that the domain differences constitute a fundamental distinction, as prior work [2] provides theoretical proof that detection and MIAs are fundamentally related, and empirical results further show that methods from each task can perform well when applied directly to the other.
> >
> > Overall, while the paper offers stronger theoretical grounding and a clearer motivation, the core conceptual idea appears largely aligned with prior work [1]. If the primary contribution lies in the theoretical analysis, it would be helpful to position this more explicitly, as it remains somewhat unclear to me whether the method introduces a fundamentally new conceptual contribution.
> >
> > **Regarding A3:**
> >
> > Thank you for the clarification. I agree that evaluating on MIMIR helps address concerns regarding temporal and lexical biases.
> >
> > However, I find some inconsistency in how WikiMIA is positioned. While the rebuttal describes it as a baseline, the paper highlights improvements on WikiMIA as a key result, for example, reporting an “AUC improvement of up to 1.5%, with its advantage being particularly pronounced in long-text scenarios” in the abstract and contributions. This suggests that WikiMIA plays a more central role in supporting the claims than implied.
> >
> > Given the known issues with WikiMIA, it would be important to more clearly reconcile this discrepancy and ensure that the main conclusions are supported by evaluations on cleaner benchmarks such as MIMIR.
> >
> > **Regarding A4:**
> >
> > Thank you for sharing the additional results. I would be particularly interested in results on more technical (low-entropy) domains included in the Pile, such as GitHub and DM Mathematics. Prior work [3] on MIMIR suggests that method behavior can differ substantially across domains, especially between high- and low-entropy settings. Evaluating on such domains would help better understand how the proposed method generalizes across different distributions.
> >
> > ---
> >
> > [1] Xu et al. TRAINING-FREE LLM-GENERATED TEXT DETECTION BY MINING TOKEN PROBABILITY SEQUENCES. ICLR 2025.
> >
> > [2] Koike et al. Machine Text Detectors are Membership Inference Attacks. CoRR 2025.
> >
> > [3] Duan et al. Do Membership Inference Attacks Work on Large Language Models? COLM 2024.

---

> > > ### Author Response · Authors · 2026-04-05
> > >
> > > Thank you for your follow-up and insightful comments. We fully understand your concerns, and we provide our detailed responses below.
> > >
> > > >**Regarding A1:**
> > >
> > > We appreciate your recognition that this paper offers stronger theoretical grounding and a clearer motivation. **We agree that the core conceptual idea of this paper is broadly similar to prior LLM-generated text detection work [1], as both analyze the volatility characteristics of token probability sequences.** Specifically, our framework is tailored to address a common limitation in existing LLM-based MIA studies, which focus on the local level but ignore global sequence volatility features (see Figure 1). Recent studies show LLM data leakage reflects deep semantic memorization, not surface word matching. Moreover, natural language text often has a very high n-gram overlap rate, such as common structures/terms [2]. **We cannot judge passage memorization by local specific words such as "is" or "the".** These common words are also naturally predicted with high probabilities, which remains a critical and memorization-specific challenge in current LLM-based MIA research.
> > >
> > > We will clarify in the revised manuscript that our primary contribution lies in the theoretical analysis, as well as clarify the comparisons with existing LLM-generated text detection work (i.e., the similarities and differences). **Although empirical observations of sequence dynamics exist in prior LLM-generated text detection work, our fundamental contribution lies in the rigorous theoretical formalization of this phenomenon for MIA tasks.** Specifically, by introducing Entropic Potential and proving Volatility Divergence (Proposition 4.3 and Lemma 4.4), AECA transforms this concept from a heuristic indicator into a framework and theoretically explains why such dynamic variations can distinguish memorization from generalized behaviors. **This further provides a rigorous theoretical foundation for this shared detection signal and complements sequence dynamics-based methods. Drawing on reference [3], we will attempt to apply this theoretical framework to text detection tasks and offer valuable insights for other related tasks.**
> > >
> > > Thanks again for your valuable comments! We hope this clarifies your concerns.
> > >
> > > >**Regarding A3:**
> > >
> > > Thank you for pointing this out. We sincerely apologize for this oversight. In the revised manuscript, in addition to presenting the superior performance of AECA on the WikiMIA benchmark, we will highlight that AECA also maintains robust performance on the cleaner MIMIR dataset, ensuring the rigor of our core conclusions.
> > >
> > > >**Regarding A4:**
> > >
> > > That's an insightful suggestion. More technical (low-entropy) domains such as code and mathematics inherently exhibit distinct fluctuation patterns, due to their fixed structures and deterministic generation logic. We hereby present evaluations on the GitHub and DM Mathematics subsets, with the results shown below:
> > >
> > > [Table A – Dataset Github]
> > > |Method|160M|1.4B|2.8B|6.9B|12B|
> > > |--|--|--|--|--|--|
> > > |PPL|0.641|0.696|0.710|0.721|0.732|
> > > |Ref|0.619|0.613|0.626|0.632|0.641|
> > > |Lowercase|0.688|0.712|0.724|0.735|0.742|
> > > |Zlib|0.685|**0.714**|0.735|0.744|0.755|
> > > |Min-K%|0.656|0.693|0.726|0.739|0.750|
> > > |Min-K%++|0.652|0.689|0.687|0.728|0.736|
> > > |DC-PDD|0.687|0.705|0.735|0.734|0.760|
> > > |PAC|0.659|0.682|0.706|0.724|0.729|
> > > |AECA|**0.699**|0.711|**0.744**|**0.756**|**0.769**|
> > >
> > > [Table B – Dataset DM Mathematics]
> > > |Method|160M|1.4B|2.8B|6.9B|12B|
> > > |--|--|--|--|--|--|
> > > |PPL|0.527|0.533|0.535|0.535|0.544|
> > > |Ref|0.462|0.475|0.481|0.472|0.465|
> > > |Lowercase|**0.558**|0.565|0.567|0.569|0.573|
> > > |Zlib|0.535|0.550|0.552|0.552|0.561|
> > > |Min-K%|0.535|0.551|0.551|0.550|0.551|
> > > |Min-K%++|0.552|0.565|**0.578**|0.579|0.573|
> > > |DC-PDD|0.546|0.542|0.554|0.556|0.561|
> > > |PAC|0.549|0.554|0.558|0.558|0.577|
> > > |AECA|**0.558**|**0.572**|0.574|**0.582**|**0.584**|
> > >
> > > As shown in the table, **AECA maintains superior overall performance even in low-entropy domains.** All corresponding subset results of MIMIR will be added to the appendix in the revised manuscript.
> > >
> > > Thank you again for your thoughtful and constructive comments, and they’ve helped improve the clarity and rigor of our work. Please feel free to continue the discussion if anything remains unclear.
> > >
> > > [1] Xu et al. TRAINING-FREE LLM-GENERATED TEXT DETECTION BY MINING TOKEN PROBABILITY SEQUENCES. ICLR 2025.
> > >
> > > [2] Duan et al. Do Membership Inference Attacks Work on Large Language Models? COLM 2024.
> > >
> > > [3] Koike et al. Machine Text Detectors are Membership Inference Attacks. CoRR 2025.

---

### Official Review · Reviewer_iDPA · 2026-03-16

**Soundness:** 3
**Presentation:** 3
**Significance:** 3
**Originality:** 3
**Overall Recommendation:** 4
**Confidence:** 3

**Summary:**

This paper introduces a new framework for detecting pretraining data in Large Language Models (LLMs), changing the focus from local token statistics to global sequence analysis. Specifically, the authors put forward Adaptive Entropic Convolutional Analysis (AECA), which regards probability sequences as dynamic signals. They add entropic potential calibration and convolutional filtering to capture the different volatility patterns caused by memorization. The paper provides theoretical proofs about volatility divergence, and the effectiveness of the method is supported by a large number of experiments on WikiMIA and MIMIR benchmarks. The numerical results show that the proposed method is efficient and better than other methods, especially in long-text situations.

**Compliance With Llm Reviewing Policy:**

Affirmed.

**Final Justification:**

Most of my concerns in the rebuttal are resolved.

**Key Questions For Authors:**

NA

**Limitations:**

Yes.

**Strengths And Weaknesses:**

**Strengths**

1.	The authors restate the task of pretraining data detection by changing the focus from local token statistics to a global sequence view, using different volatility patterns to tell mechanical memorization apart from reasoning-based generation.
2.	The authors put forward Adaptive Entropic Convolutional Analysis (AECA), a framework that combines self-information calibration with convolutional filtering to effectively capture global volatility patterns that local statistical methods fail to notice.
3.	A large number of experiments on WikiMIA and MIMIR benchmarks show that the proposed method is better than the most advanced baselines by up to 1.5% in AUC, while keeping computational efficiency similar to simple perplexity baselines.

**Weaknesses**

1.	While the paper establishes a theoretical lower bound for $\lambda$ in Lemma 4.4, its empirical selection relies solely on grid search. We recommend that the authors supplement a thorough ablation study on $\lambda$ to validate its influence. In addition, to better illustrate the roles of $ \sigma(S)$ and $\sigma(L)$, it would be helpful if the authors could add a 2D visualization of the joint distribution of memorized vs. non-memorized samples using these two metrics as axes in the revised version. This visualization can help clarify how these two indicators distinguish the two sample groups.
2.	The method uses a subset of the C4 dataset with Laplace smoothing to estimate token frequencies. How sensitive is AECA's performance to (a) the scale of the reference corpus, (b) its domain alignment with the target detection task, and (c) the smoothing coefficient $\alpha$? A systematic ablation on these factors would strengthen the practical guidance for deploying AECA in scenarios where reference data availability varies.
3.	The Entropic Difference Response adopts a first-order difference kernel k=[1,−1] as a high-pass filter. Have the authors experimented with alternative kernels (e.g., Sobel, Laplacian, or learnable filters) or other volatility metrics (e.g., wavelet coefficients, entropy rate, or higher-order statistical moments)? Clarifying whether the current design is empirically optimal or primarily motivated by theoretical simplicity would help assess the method's flexibility and potential for further improvement.
4.	The core claim that memorized sequences exhibit distinct volatility patterns compared to reasoning‑driven generation is well‑supported. Can this framework be extended to related detection tasks—for instance, distinguishing genuine reasoning from superficial pattern matching in chain‑of‑thought outputs from reasoning‑enhanced large language models[A], or differentiating hallucinatory content from factually grounded generations[B]? If the authors have early insights or constraints regarding these extensions, sharing them would significantly enhance the broader impact of this work.

**Reference**

[A] Efficient Reasoning Models: A Survey. 2025 TMLR

[B] A Survey of Multimodal Hallucination Evaluation and Detection. 2025 IJCV

---

> ### Author Rebuttal · Authors · 2026-03-27
>
> We sincerely thank you for your positive feedback on the motivation, core idea, and experimental results of our paper. Below is our point-by-point response to your questions.
>
> >**Q1:** Supplement an ablation study on $\lambda$ to validate its influence.
>
> **A1:** Thank you for this constructive suggestion. For completeness, we conducted additional ablation experiments on WikiMIA with GPT-Neo-2.7B and Pythia-12B, setting $\lambda$ to 0, 0.01, 0.1, 1–10. Results below:
>
> |$\lambda$|0|0.01|0.1|1|2|3|4|5|6|7|8|9|10|
> |-|-|-|-|-|-|-|-|-|-|-|-|-|-|
> |GPT-Neo-2.7B|0.599|0.600|0.606|0.662|0.689|0.697|0.697|0.698|0.698|0.692|0.690|0.688|0.687|
> |Pythia-12B|0.582|0.583|0.592|0.662|0.699|0.712|0.715|0.716|0.717|0.716|0.715|0.715|0.714|
>
> As shown, optimal $\lambda$ varies **slightly** across models but is around 5. Setting $\lambda$ too small or omitting it degrades performance significantly, **empirically validating Lemma 4.4 ($\lambda$ cannot be below its theoretical lower bound).** We will add these results in the revised paper. Thank you for the feedback!
>
> >**Q2:** Add a 2D visualization of the joint distribution of memorized vs. non-memorized samples using $\sigma(S)$ and $\sigma(L)$ as axes in the revised version.
>
> **A2:** Thank you for this valuable suggestion. We conducted experiments on WikiMIA with Pythia-12B, computing average $\sigma(S)$ and $\sigma(L)$ for memorized/non-memorized samples. Numerical results below; visualization will be added to the revised paper.
>
> |Group|Memorized|non-Memorized|
> |-|-|-|
> |$\sigma(S)$|2.870|2.635|
> |$\sigma(L)$|1.996|2.215|
> |$\Delta$|0.874|0.420|
>
> As shown, our method amplifies the fluctuation difference of memorized samples, enabling better separation of the two sample types.
>
> >**Q3:** How sensitive is AECA's performance to (a) the scale of the reference corpus, (b) its domain alignment with the target detection task, and (c) the smoothing coefficient?
>
> **A3:** Thank you for pointing this out. To verify AECA’s robustness: (a&b) $\approx$ 1GB/10GB of C4 and Case-law [1] (legal domain) as reference corpora; (c) smoothing coefficient $\alpha$ tested at 0.01, 0.05, 0.1, 0.2, 0.5, 1.0, 2.0, 3.0. All experiments were on WikiMIA with Pythia-12B. Results below:
>
> [Table A – Reference Corpora $D_{\text{ref}}$]
> |$D_{\text{ref}}$|C4 $\approx$ 1GB|C4 $\approx$ 10GB|Case-law $\approx$ 1GB|Case-law $\approx$ 10GB|
> |-|-|-|-|-|
> |Pythia-12B|0.711|0.716|0.705|0.713|
>
> [Table B – Smoothing Factor $\alpha$]
> |$\alpha$|0.01|0.05|0.1|0.2|0.5|1.0|2.0|3.0|
> |-|-|-|-|-|-|-|-|-|
> |Pythia-12B|0.716|0.716|0.717|0.716|0.717|0.716|0.716|0.716|
>
> As shown, **AECA’s performance has no significant variance across reference corpora and $\alpha$.**
>
> >**Q4:** Have the authors experimented with alternative kernels (e.g., Sobel, Laplacian, or learnable filters) or other volatility metrics (e.g., wavelet coefficients, entropy rate, or higher-order statistical moments)?
>
> **A4:** We appreciate your constructive suggestions. We previously tested other kernels (e.g., Laplacian) and volatility metrics (e.g., 3rd-order Moments). Following your advice, we added experiments on WikiMIA with Pythia-12B. Results below:
>
> [Table A – Kernel]
> |Kernel|Sobel|Laplacian|Ours|
> |-|-|-|-|
> |Pythia-12B|0.708|0.699|**0.716**|
>
> [Table B – Volatility Metric]
> |Volatility Metric|Wavelet Coefficients|Entropy Rate|3rd-order Moments|4th-order Moments|Ours|
> |-|-|-|-|-|-|
> |Pythia-12B|0.680|0.548|0.542|0.559|**0.716**|
>
> As shown, other kernels/metrics are effective, but our first-order difference kernel + standard deviation **achieves optimal performance with simplicity and low computational cost. Their favorable mathematical properties (e.g., Popoviciu's inequality for standard deviation) support Lemma 4.4.**
>
> >**Q5:** Can this framework be extended to related detection tasks—for instance, effective reasoning and hallucination detection?
>
> **A5:** Yes, this framework can be extended to these related detection tasks. We have done preliminary research on these extensions and are glad to share our thoughts.
>
> For CoT reasoning, LLM reasoning quality can be assessed via token confidence dynamics during generation, offering an entropy dynamics view of reasoning [2]. For hallucination detection, entropy-based uncertainty is a strong indicator: high semantic entropy implies uncertainty/hallucinations, while low entropy indicates factual generation [3].
>
> These mechanisms share a similar core motivation with AECA. **We plan to extend our method to these tasks and add relevant discussions to the Future Work section** in the revised manuscript. Thank you for highlighting this promising direction.
>
> Please feel free to contact us if you require further clarification.
>
> Reference:
>
> [1] https://huggingface.co/datasets/HFforLegal/case-law
>
> [2] Zhu, Chenghua, et al. EDIS: Diagnosing LLM Reasoning via Entropy Dynamics. arXiv preprint arXiv:2602.01288 (2026).
>
> [3] Farquhar, Sebastian, et al. Detecting hallucinations in large language models using semantic entropy. Nature 2024.

---

> > ### Author Rebuttal · Reviewer_iDPA · 2026-04-04
> >
> > Most of my concerns are resolved. I have decided to maintain the original score.

---

> > > ### Author Response · Authors · 2026-04-04
> > >
> > > Thank you so much for your encouraging feedback. We are glad our responses have resolved most of your concerns. We will fully incorporate all the discussed changes into the revised manuscript. Your support truly means a lot to us!

---

### Decision · Program_Chairs · 2026-04-30

**Decision:**

Accept (regular)

**Comment:**

This paper addresses pretraining data detection for LLMs and introduces AECA, which moves beyond local token-level signals and instead models global volatility patterns in token probability sequences. Overall, reviewers found the paper technically sound, well motivated, and supported by a fairly broad experimental study across models, scales, and benchmarks. The theoretical perspective around volatility divergence was also viewed as a real strength.

The main points of discussion were novelty, the assumptions underlying the theory, and how robust the empirical evidence is. In the rebuttal and subsequent discussion, the authors engaged seriously with these concerns. They clarified how the paper differs from prior work, provided additional evidence on robustness and statistical significance, and responded to questions about benchmark selection and generality. These replies addressed a substantial part of the reviewers’ concerns and increased confidence in the paper.

One reviewer still remained unconvinced that the paper is conceptually distinct enough from prior sequence-based approaches, and would have preferred stronger evidence on cleaner or more diverse settings. That is a fair reservation. Still, looking at the discussion as a whole, the paper appears to make a meaningful contribution, with solid technical quality and enough empirical support to justify acceptance. I therefore recommend weak acceptance.